# Arabidopsis HAK5 under low K$^+$ availability operates as PMF powered high-affinity K$^+$ transporter

Tobias Maierhofer [1,7] ✉, Sönke Scherzer [1,2,7], Armando Carpaneto [3,4] ✉, Thomas D. Müller[1], Jose M. Pardo [5], Inga Hänelt [6], Dietmar Geiger[1] & Rainer Hedrich[1]

Plants can survive in soils of low micromolar potassium (K$^+$) concentrations. Root K$^+$ intake is accomplished by the K$^+$ channel AKT1 and KUP/HAK/KT type high-affinity K$^+$ transporters. Arabidopsis HAK5 mutants impaired in low K$^+$ acquisition have been identified already more than two decades ago, the molecular mechanism, however, is still a matter of debate also because of lack of direct measurements of HAK5-mediated K$^+$ currents. When we expressed AtHAK5 in Xenopus oocytes together with CBL1/CIPK23, no inward currents were elicited in sufficient K$^+$ media. Under low K$^+$ and inward-directed proton motive force (PMF), the inward K$^+$ current increased indicating that HAK5 energetically couples the uphill transport of K$^+$ to the downhill flux of H$^+$. At extracellular K$^+$ concentrations above 25 µM, the initial rise in current was followed by a concentration-graded inactivation. When we replaced Tyr450 in AtHAK5 to Ala the K$^+$ affinity strongly decreased, indicating that AtHAK5 position Y450 holds a key for K$^+$ sensing and transport. When the soil K$^+$ concentration drops toward the range that thermodynamically cannot be covered by AKT1, the AtHAK5 K$^+$/H$^+$ symporter progressively takes over K$^+$ nutrition. Therefore, optimizing K$^+$ use efficiency of crops, HAK5 could be key for low K$^+$ tolerant agriculture.

Potassium represents the major inorganic cation required for turgor and membrane potential formation[1,2]. Because potassium is an essential macronutrient, plants need to adjust their K$^+$ uptake systems to meet their demands on soils with widely varying nutrient sources. The K$^+$ concentration in the cytoplasm of root cells is maintained constant around 100 mM. In the depletion zone around the roots, however, the K$^+$ concentration can drop from the millimolar to the low micromolar range[3]. Thus, K$^+$ gradients between the cytoplasm and the external solution of up to 4-5 orders of magnitude are not uncommon under

natural conditions. Root potassium uptake has been described in classical tracer experiments as a biphasic process[4] characterized by a high-affinity system with apparent affinities of approximately 20 µM K$^+$ and a low-affinity system operating in the range between 200 µM and 2 mM K$^+$.

Genetic disruption of AKT1 (*akt1-1*) impairs K$^+$ uptake into Arabidopsis roots and seedling growth on low K$^+$ media ([K$^+$] < 1 mM)[5,6]. AKT1 is a voltage-dependent (inward-rectifying) member of the plant Shaker K$^+$ channel family[6,7]. Uptake kinetics of the K$^+$ channel mutant

[1]Molecular Plant Physiology and Biophysics, Julius-von-Sachs Institute, Biocenter, Julius-Maximilians-Universität Würzburg, Würzburg 97082, Germany. [2]Institute of Education and Student Affairs, University of Münster, Münster, Germany. [3]Department of Earth, Environment and Life Sciences (DISTAV), University of Genova, Genova, Italy. [4]Institute of Biophysics, National Research Council, Genova, Italy. [5]Instituto de Bioquimica Vegetal y Fotosintesis (IBVF), CSIC-Universidad de Sevilla, Sevilla, Spain. [6]Institute of Biochemistry, Goethe University Frankfurt, Frankfurt am Main, Germany. [7]These authors contributed equally: Tobias Maierhofer, Sönke Scherzer. ✉e-mail: t.maierhofer@botanik.uni-wuerzburg.de; armando.carpaneto@unige.it

akt1-1 showed that root K⁺ uptake is limited in the intermediate to low K⁺ affinity range. In contrast, AtHAK5, a potassium transporter from the KUP/HAK/KT family[8,9], is operating in the high-affinity range. The protein kinase CIPK23 seems key for both the low and high affinity K⁺ uptake[10–12]. The transcription of calcium sensory CBL-interacting protein kinase CIPK23 is induced via the transcription factor STOP1 at low K⁺[13]. In turn, CBL1/CIPK23 can activate both AKT1 and HAK5[11,14].

Early studies quantified the K⁺ root uptake using Rb⁺ ions as a tracer[5,15,16] since HAK/KUP-derived K⁺ currents could not be monitored directly[9]. The apparent contribution of AKT1 and HAK5 to total K⁺ tracer Rb⁺ uptake was, however, approximated using NH₄⁺ as discriminator[15]. On media with sufficient K⁺, the AKT1 loss-of-function mutant showed normal growth in the presence of high NH₄⁺, while NH₄⁺ suppresses *akt1-1* growth in low K⁺[6]. However, some K⁺ channels do not transport Rb⁺[17,18] and NH₄⁺ is a substrate of K⁺ channels and ammonium transporters[19,20]. In addition, AKT1 and HAK5 for activation compete for the same protein kinase[10–12]. Given these imponderables, HAK5 properties derived from experiments with whole Arabidopsis roots can provide only rough estimates about the molecular transport mechanism.

To analyze the biophysical properties of AtHAK5 directly, we expressed the Arabidopsis transporter in Xenopus oocytes for electrophysiological studies. Voltage-clamp experiments uncovered the hallmark AtHAK5 characteristics of a voltage-dependent, H⁺-coupled, high-affinity K⁺ uptake system.

## Results

### AtHAK5 is turned on when extracellular K⁺ levels become limiting

The Xenopus system has been proven a reliable and robust heterologous expression system for functional testing of animal and plant membrane transporters[21–23]. The combination of Two Electrode Voltage-Clamp (TEVC) technique and K⁺ selective electrodes enables simultaneous recordings of the transport activity in the context of the apparent extracellular K⁺ concentrations. When AKT1 was co-expressed and activated by the CBL1/CIPK23 calcium sensor kinase pair, in the presence of 2 mM K⁺ and stimulated by membrane hyperpolarization macroscopic inward K⁺ currents appeared (Fig. 1A). During the replacement of 2 mM K⁺ by media containing 20 μM K⁺ only, currents dropped. The K⁺ currents remaining, however, did not vanish when K⁺ was lowered to nominally zero K⁺. This indicates that AKT1 does not recognize K⁺ moieties in the low micromolar range.

Like AKT1, AtHAK5 transport activity strongly depends on its co-expression with CBL1 and CIPK23, since inward K⁺ currents increased tenfold in the presence of the Ca²⁺sensor kinase pair compared to the current amplitudes observed for AtHAK5 expressed alone

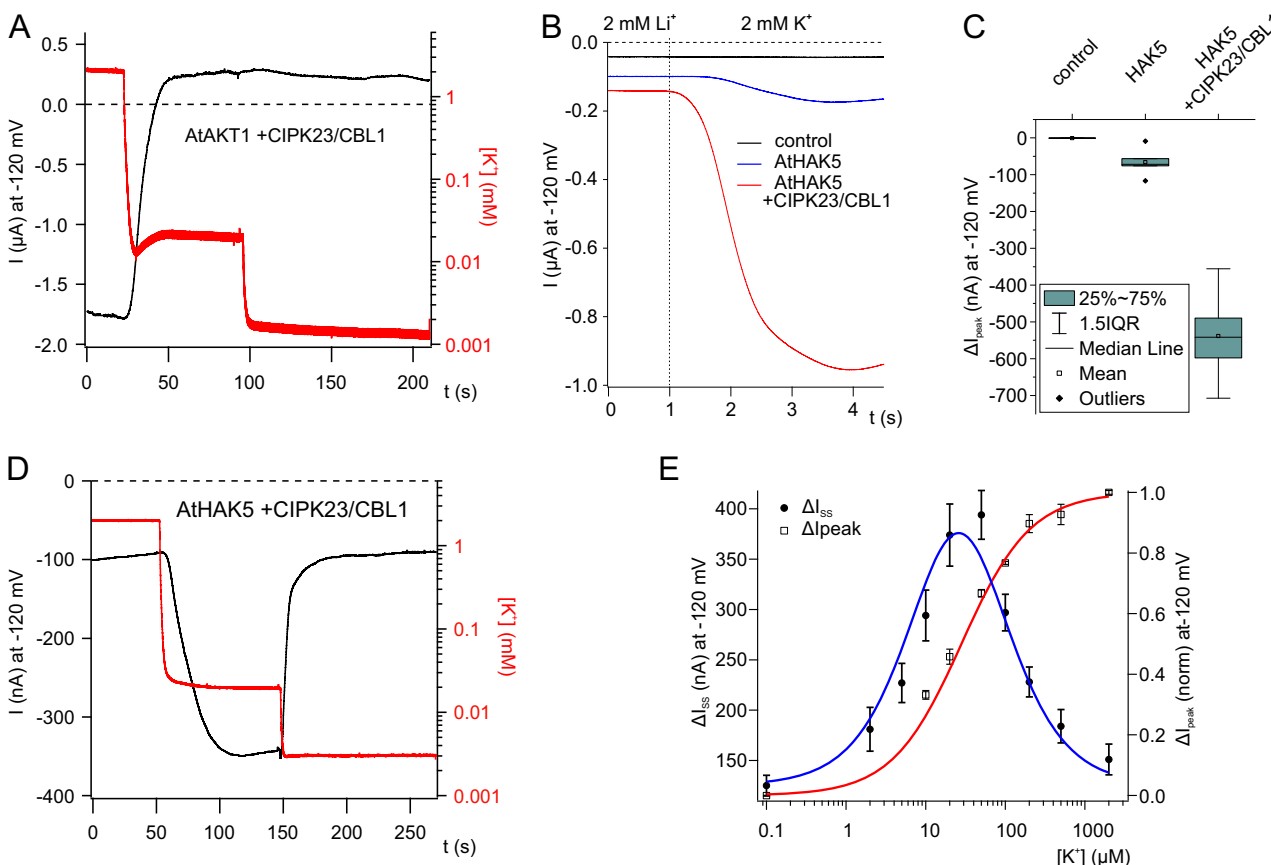

**Fig. 1 | Potassium dependent activation of AtHAK5. A** Representative original currents (black trace) at −120 mV of an oocyte co-expressing AtAKT1 and CIPK23/CBL1 at pH 4.5 in the presence of different potassium concentrations (2 mM, 20 μM or nominally 0 μM K⁺). Simultaneously, the bath K⁺ concentration was recorded via K⁺-selective electrodes (representative trace in red). **B** Representative K⁺-induced currents at −120 mV of either control oocytes or oocytes expressing either AtHAK5 alone or co-expressing AtHAK5 together with CIPK23/CBL1. **C** Quantification of K⁺-induced peak currents at −120 mV of either control oocytes (*n* = 4 experiments) or oocytes expressing AtHAK5 in the presence or absence of CIPK23/CBL1 (*n* = 5 experiments). **D** Measurement of original current traces (in black) at −120 mV in oocytes co-expressing AtHAK5 and CIPK23/CBL1 at pH 4.5 in the presence of different potassium concentrations (2 mM, 20 μM or nominally 0 μM K⁺). In the bath a K⁺-selective electrode simultaneously recorded the apparent K⁺ concentration (red trace; Representative traces are shown). **E** Normalized whole-oocyte K⁺-induced peak currents (ΔI_peak) (*n* = 3 experiments, mean ± SD) or steady-state currents (ΔI_SS) (*n* = 4 experiments, mean ± SEM) at −120 mV at pH4.5 are plotted against the applied K⁺-concentration. K_m (K⁺) was calculated by fitting ΔI_peak with a Michaelis-Menten equation. The modified Michaelis-Menten function used to fit ΔI_SS is described in the methods section.

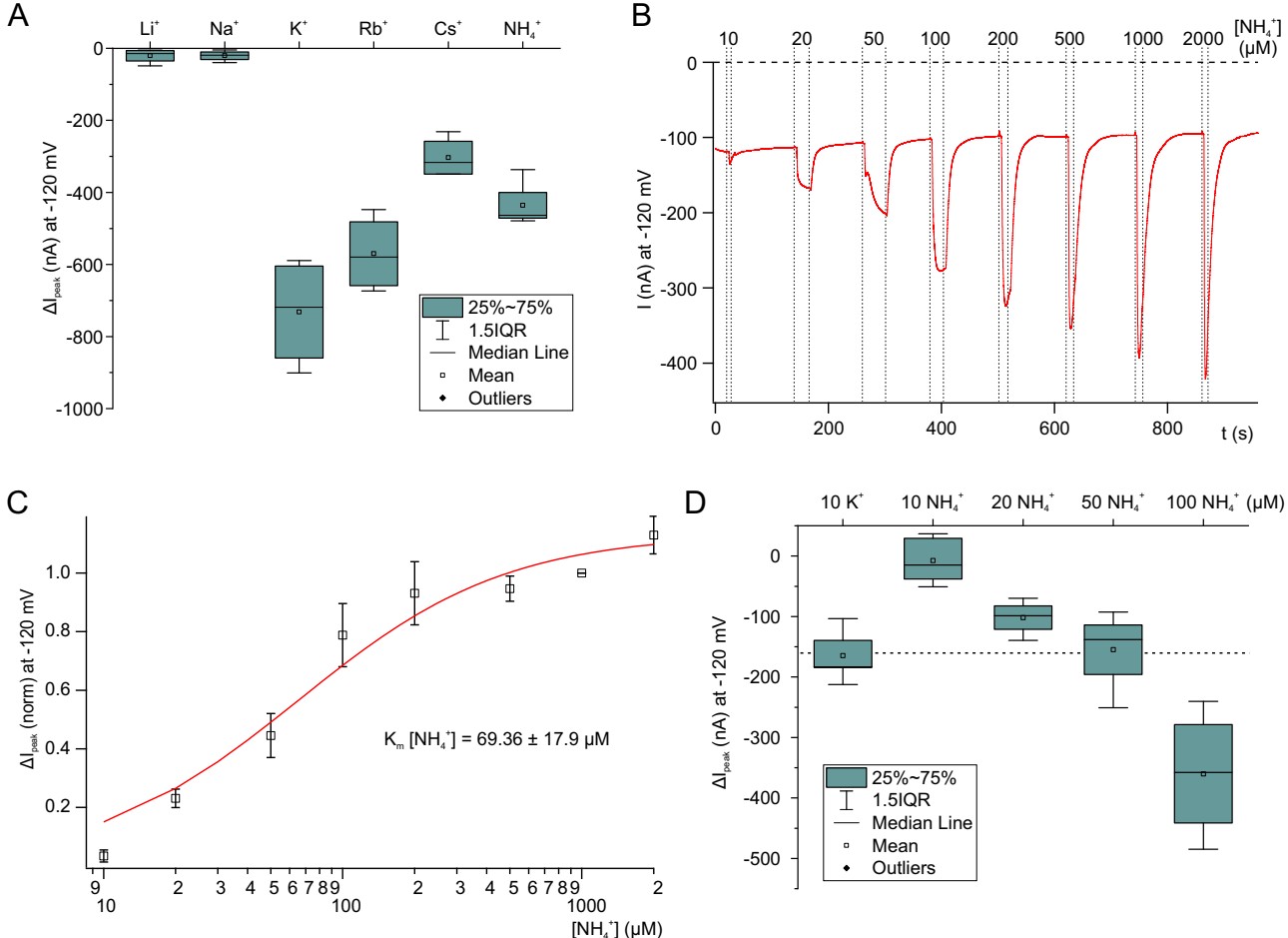

**Fig. 2 | Cation dependency of AtHAK5. A** Box plot of cation-induced peak currents ($\Delta I_{peak}$) at −120 mV of oocytes co-expressing AtHAK5 and CIPK23/CBL1 in response to either 2 mM Li$^+$, Na$^+$, K$^+$, Rb$^+$, Cs$^+$ or NH$_4^+$ ($n = 5$ experiments). **B** Representative current response of AtHAK5/CIPK23/CBL1 co-expressing oocytes upon application of different NH$_4^+$ concentrations. **C** Normalized whole-oocyte NH$_4^+$-induced peak currents ($\Delta I_{peak}$) at −120 mV (pH4.5) plotted against the applied NH$_4^+$- concentration. $K_m$ (NH$_4^+$) was calculated by fitting $\Delta I_{peak}$ with a Michaelis-Menten equation. ($n = 6$ experiments, mean ± SD). **D** Box plot of peak current response ($\Delta I_{peak}$) of AtHAK5/CIPK23/CBL1 co-expressing oocytes in the presence of either 10 µM K$^+$ or different NH$_4^+$-concentrations as indicated in the figure ($n = 4$ experiments for 20, 50, 100 µM NH$_4^+$ and $n = 5$ experiments for 10 µM K$^+$ and NH$_4^+$, mean ± SD).

(Fig. 1B, C, c.f[10].). When K$^+$ concentration shift experiments (cf. Figure 1A) were performed in the presence of AtHAK5, the situation changed significantly. In oocytes co-expressing AtHAK5 with CBL1/CIPK23 that were exposed to 2 mM K$^+$ no transport activity could be recorded (Fig. 1D). In contrast to AKT1, the shift to 20 µM K$^+$ elicited HAK5-mediated inward currents that disappeared in K$^+$ free media (Fig. 1D). When shifting from 2 mM K$^+$ to low K$^+$ concentrations between 10 and 200 µM, AtHAK5 activated in a concentration dependent manner (Supplementary Fig. 1A). Plotting the low K$^+$ induced currents as a function of external [K$^+$] resulted in a bell-shaped curve with highest AtHAK5 activity at about 20 µM K$^+$ that decreased with increasing K$^+$ (Supplementary Fig. 1A and B). This behavior indicates that AtHAK5 is inactive at sufficient external K$^+$ but is turned on in the low K$^+$ range.

When the K$^+$ concentration was altered stepwise from nominally zero to either 10, 20, 50, 100, 200, 500, 1000 or 2000 µM K$^+$, the current increased in a K$^+$-dependent manner (Fig. 1E). When the peak K$^+$ currents were plotted as function of the K$^+$ concentration, the saturating dose-response curve could be fitted with a Michaelis-Menten function yielding a $K_m$ value of 23.56 µM ( ± 1.3 µM) (Fig. 1E).

**AtHAK5 transport is sensitive to NH$_4^+$**
To study the monovalent cation selectivity of HAK5, K$^+$ was replaced by the same concentration of Na$^+$, Li$^+$, Rb$^+$, and Cs$^+$. In the presence of Rb$^+$,

currents were comparable to those elicited by K$^+$. Cs$^+$-evoked currents were about 51.1 ± 4.4 % smaller than K$^+$-induced ones (Fig. 2A, Supplementary Fig. 2A). Dose response curves for Rb$^+$ and Cs$^+$ revealed $K_m$ values of 22.96 ± 7.1 µM and 18.90 ± 12.9 µM, respectively (Supplementary Fig. 2B, C). During the shift from Li$^+$ to Na$^+$, however, no inward currents were triggered (Fig. 2A, Supplementary Fig. 2A).

When challenging AtHAK5 under K$^+$-free conditions with increasing concentrations of ammonium from 10 to 2000 µM, NH$_4^+$ triggered inward currents in a dose-dependent manner. This fact documents that besides K$^+$, the nitrogen nutrient NH$_4^+$ represents a physiologically relevant AtHAK5 substrate as well (Fig. 2B). NH$_4^+$-evoked currents were about 37.6 ± 9.6 % smaller than K$^+$-induced ones (Fig. 2A, Supplementary Fig. 2A) and of lower affinity ($K_m$ 69.36 ± 17.9 µM, Fig. 2C). To produce comparable inward current amplitudes with ammonium as with 10 µM K$^+$, AtHAK5 had to be challenged with an about 5-times higher NH$_4^+$ concentration (Fig. 2D, Supplementary Fig. 2D), supporting our finding that AtHAK5 transports K$^+$ at higher affinity than NH$_4^+$ (cf. Figure 1E and Fig. 2C). Given that high affinity (3 µM) AMT-type ammonium transporters are also expressed in the root[24], AtHAK5 predominantly seems to function as K$^+$ transporter in planta. At saturating NH$_4^+$ concentrations (1 mM; Supplementary Fig. 2E), however, the addition of 10 or 100 µM K$^+$ did not further increase AtHAK5 inward currents. This finding is in line with high 1 mM NH$_4^+$

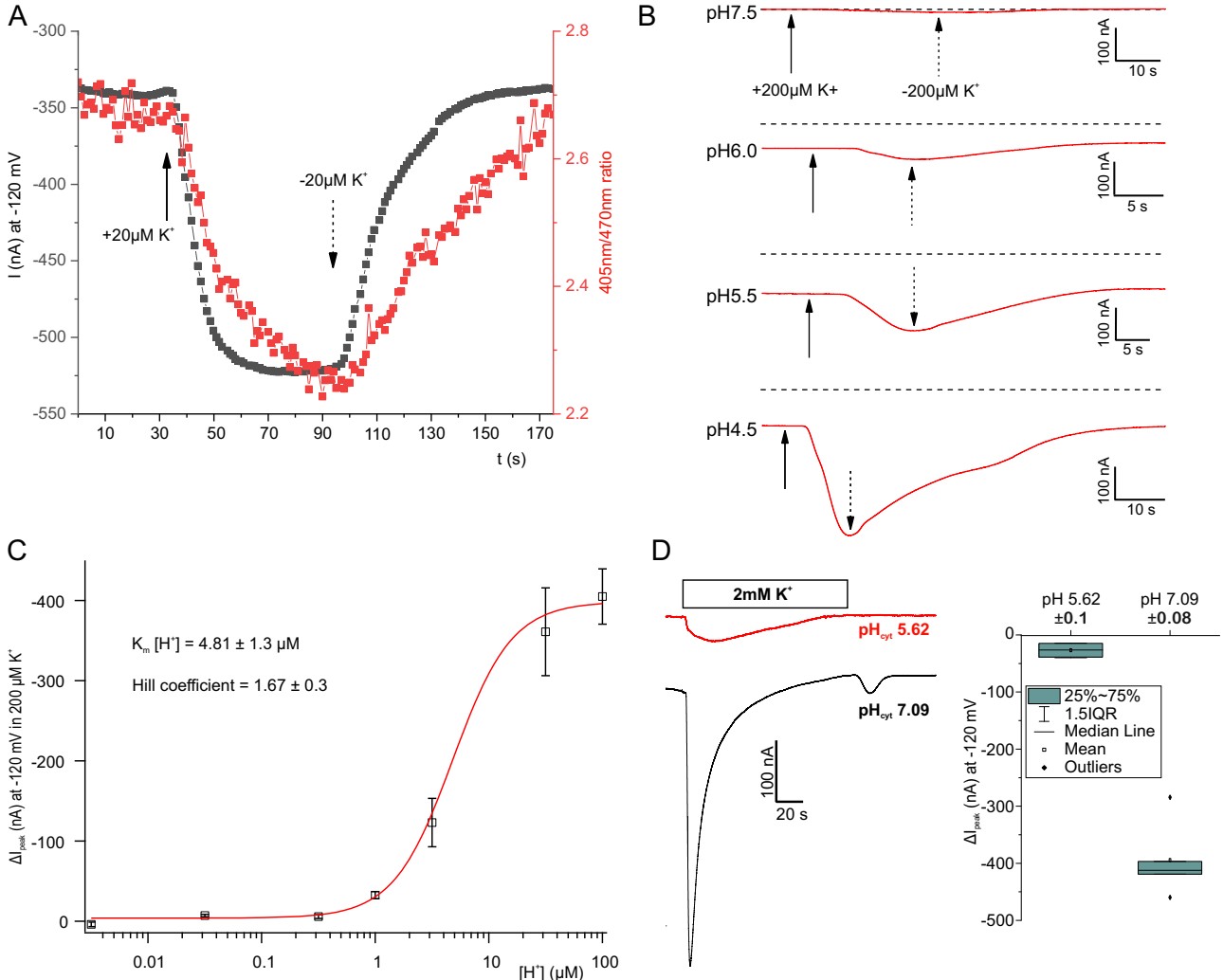

**Fig. 3 | pH dependency of AtHAK5. A** Simultaneous recording of current response (black trace) and cytosolic pH changes (red trace) in oocytes expressing pHluorin:HAK5 with CIPK23/CBL1. Oocytes were clamped to −120 mV and currents were triggered by perfusion with 20 μM K⁺. A drop in 405 nm/470 nm ratio represents cytosolic acidification. Representative measurement from 4 independent experiments is shown. **B** Representative current response of AtHAK5/CIPK23/CBL1 co-expressing oocytes upon application of 200 μM K⁺ at different pH (as indicated in the figure). **C** Whole-oocyte K⁺-induced peak currents ($\Delta I_{peak}$) at −120 mV plotted against the applied H⁺-concentration. $K_m$ (H⁺) was calculated by fitting $\Delta I_{peak}$ with a Hill equation. (*n* = 4 experiments for pH 8.5, 7.5, 6 and 4, *n* = 8 experiments for pH 6.5, 5.5 and 4.5, mean ± SD). **D** Left panel: Representative current response of AtHAK5/CIPK23/CBL1 co-expressing oocytes at pH$_{ext}$ 4.5 upon application of 2 mM K⁺ in the presence (red) or absence (black) of cytosolic acidification via sodium acetate treatment. pH$_{cyt}$ was recorded via H⁺ selective electrodes. Right panel: Box plot of peak current responses upon application of 2 mM K⁺ at either pH$_{cyt}$ 5.62 ± 0.1 (*n* = 4 experiments, mean ± SD) or pH$_{cyt}$ 7.09 ± 0.08 (*n* = 5 experiments, mean ± SD).

concentrations outcompeting HAK5-mediated K⁺ uptake by Arabidopsis roots[15,25,26].

## HAK5 is a proton-driven K⁺ transporter

Plant high-affinity HAK5-like transporters are supposed to mediate K⁺/H⁺ symport[10,17] as it could be demonstrated for KimA from *Bacillus subtilis*[27]. To test whether AtHAK5 represents a H⁺ and K⁺ symporter, we fused the pH-sensitive reporter pHluorin[28] to the N-terminus of AtHAK5. pHluorin:HAK5-expressing oocytes (Supplementary Fig. 3A) were clamped to −120mV and perfused with 20 μM KCl pH4.5 (Fig. 3A). In line with the function of a H⁺ and K⁺ symporter, the inward currents were tightly coupled to the acidification of the cytoplasmic face of the plasma membrane, visualized by a decrease of pHluorin fluorescence ratio (405/470). During the K⁺ wash-out phase, the ion currents and the cytosolic pH returned to their pre-stimulus settings (Fig. 3A).

Under certain conditions, root proton secretion can acidify the rhizosphere by more than two pH units[29,30]. Such an acidification will increase the proton-motive-force (PMF) for H⁺/solute transport

manyfold. To study the impact of pH changes on AtHAK5 mediated K⁺/H⁺ symport, we exposed HAK5-expressing oocytes to 200 μM K⁺ while varying the H⁺ concentration stepwise from pH 8.5 to 4.0 (Fig. 3B, C). At −120 mV, the drop in pH resulted in a H⁺ dose-dependent increase in inward current. The current amplitudes plotted as a function of the H⁺ concentration could be fitted with a Hill equation, with a half-maximal AtHAK5 activity at 4.81 (± 1.3) μM H⁺ (Fig. 3C). This transport characteristic of AtHAK5 was found independent of the external K⁺ concentration. When performing the same experiment in the presence of 20 μM K⁺, the half-maximal pH-dependent AtHAK5 activity (3.15 ± 0.8 μM H⁺) (Supplementary Fig. 3B, C) was found similar as in the presence of 200 μM K⁺ (cf. Fig. 3C).

How does the cytosolic pH influence the transport capacity of AtHAK5? Using H⁺-sensitive microelectrodes, we determined a pH of around 7.1 in the oocyte's cytosol (pH$_{cyt}$) (Fig. 3D, c.f[23].). To decrease the cytosolic pH of AtHAK5-expressing oocytes and thereby vanishing the proton gradient across the plasma membrane, we incubated oocytes in 30 mM sodium acetate for 10 min and measured a stable

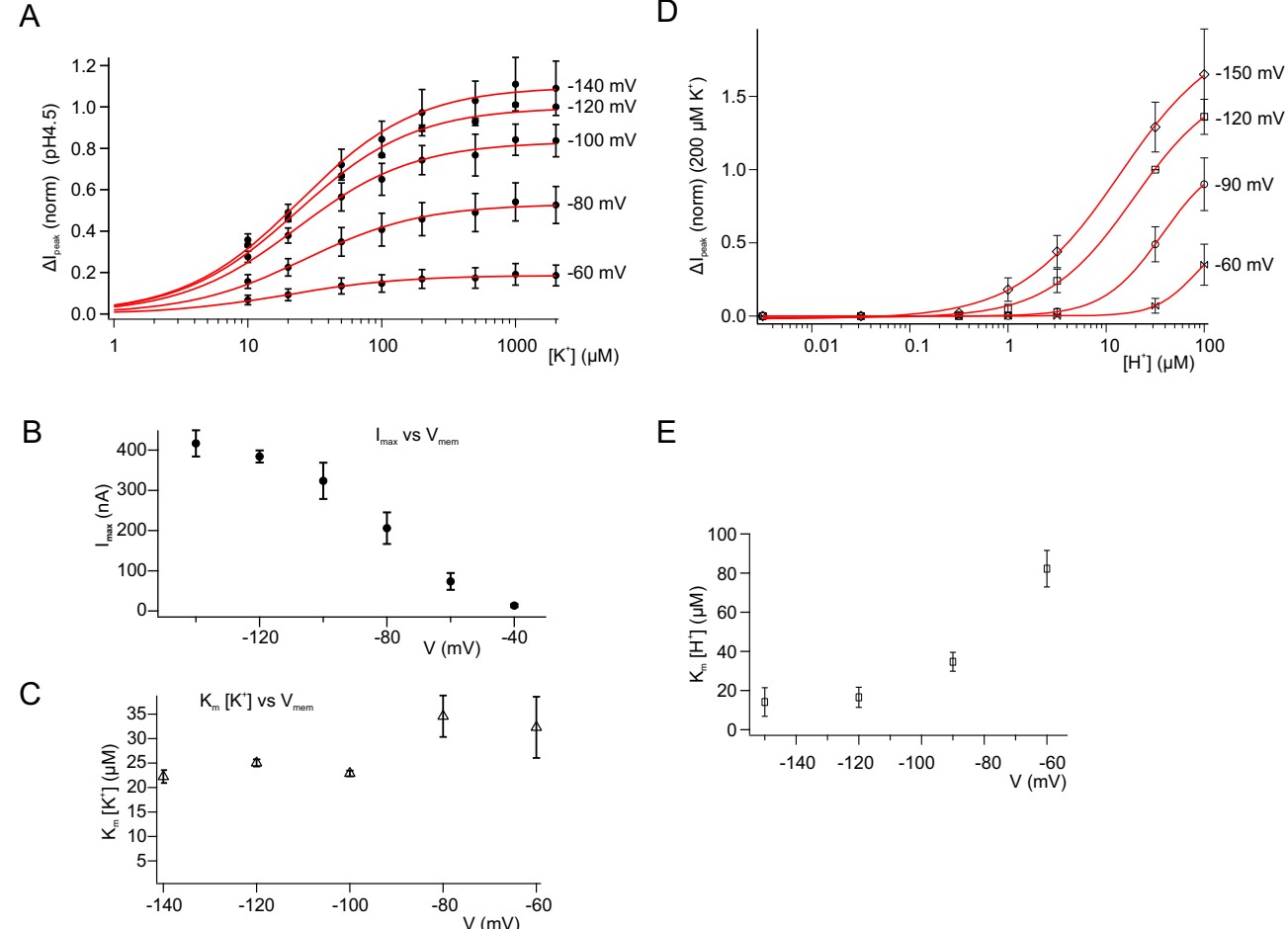

**Fig. 4 | Voltage dependency of AtHAK5. A** Normalized whole-oocyte currents of AtHAK5 and CIPK23/CBL1 co-expressing oocytes. K⁺-induced peak currents ($\Delta I_{peak}$) at different voltages (as indicated in the figure) (pH4.5) were plotted against the applied K⁺-concentration. Data points were fitted with a Michaelis-Menten equation. ($n = 4$ experiments, mean ± SEM). **B** Maximum currents ($I_{max}$) derived from the Michaelis-Menten fits shown in A) were plotted against the applied voltage ($n = 4$ experiments, mean ± SEM). **C)** $K_m$ (K⁺) values derived from the Michaelis-Menten fits shown in A) were plotted as a function of the applied voltage ($n = 4$ experiments,

mean ± SEM). **D** Normalized whole-oocyte currents of AtHAK5 and CIPK23/CBL1 co-expressing oocytes. Peak currents ($\Delta Ipeak$) were induced with 200 μM K⁺ at different voltages (as indicated in the figure) and plotted against the applied H⁺-concentration. Data points were fitted with a Hill equation ($n = 5$ experiments for −150 mV and $n = 7$ experiments for −60, −90, and −120 mV, mean ± SD). **E)** $K_m$ [H⁺] values derived from the data shown in D) were plotted against the applied voltage (mean ± SD).

pH$_{cyt}$ of 5.6 (Fig. 3D, c.f[31].). While keeping the external H⁺ concentration constant at pH 4.5, the amplitude of K⁺-induced AtHAK5 currents of acetate pre-treated oocytes reached a fraction of the amplitude measured in controls only (Fig. 3D). This behavior underlines that the proton gradient across the plasma membrane drives the K⁺/H⁺ symport of AtHAK5-expressing oocytes.

## AtHAK5 is activated by membrane hyperpolarization

To study the contribution of the voltage to the AtHAK5-mediated K⁺ transport, we stepped the membrane potential of the oocyte from −60 to −140 mV in 20 mV decrements and varied the K⁺ concentrations from 10 μM to 2 mM (Fig. 4A). At a hyperpolarized potential of −140 mV a maximal inward current ($I_{max}$) of up to 417 ± 65 nA was calculated by fitting the data in Fig. 4A with a Michaelis-Menten function. Upon depolarization, $I_{max}$ dropped to cease completely at −40 mV (Fig. 4B). This voltage dependency is reminiscent to hyperpolarization-activated plant inward-rectifying K⁺ channels such as KAT1 and AKT1[7,32].

Given that the electrophoretic force of both K⁺ and H⁺ increases with negative-going membrane potentials, we analyzed the K⁺- and H⁺-dependent AtHAK5 currents independently. When plotted as a function of the membrane potential, the $K_m$ [K⁺] deduced from data in

Fig. 4A was weakly voltage dependent only (Fig. 4C). This indicates that the K⁺ binding site in the high-affinity K⁺/H⁺ symporter is not sensing the drop in the transmembrane potential difference. To answer questions about H⁺ binding, we kept the K⁺ concentration at 200 μM, and monitored the AtHAK5 currents at −150, −120, −90 and −60 mV as a function of the external [H⁺] (Fig. 4D). Fitting the saturation curves with a Michaelis Menten equation revealed a pronounced voltage-dependent shift of $K_m$ [H⁺] to lower H⁺ concentrations with more negative membrane potentials (Fig. 4D). A + 30 mV depolarization (−120 mV to −90 mV) caused a drop in $K_m$ [H⁺] of about 20 μM (Fig. 4E). This documents that in contrast to the transporter's K⁺ binding site(s), the binding site(s) for protons sense(s) the electrical field. Transport proteins just like all enzymes must be viewed as nanomachines that obey the rules of thermodynamics. To describe and to better understand the differences in voltage dependence of potassium and proton currents and affinities (Fig. 4A–E), we developed a four-state kinetic scheme (Supplementary Fig. 4A–D; for more details see Methods section). The resulting simulations (Supplementary Fig. 4E–I) were found well in agreement with the respective experimental data of Fig. 4A-E suggesting that in the ground state (C1 state) negatively charged amino acids face the extracellular side of the membrane in the

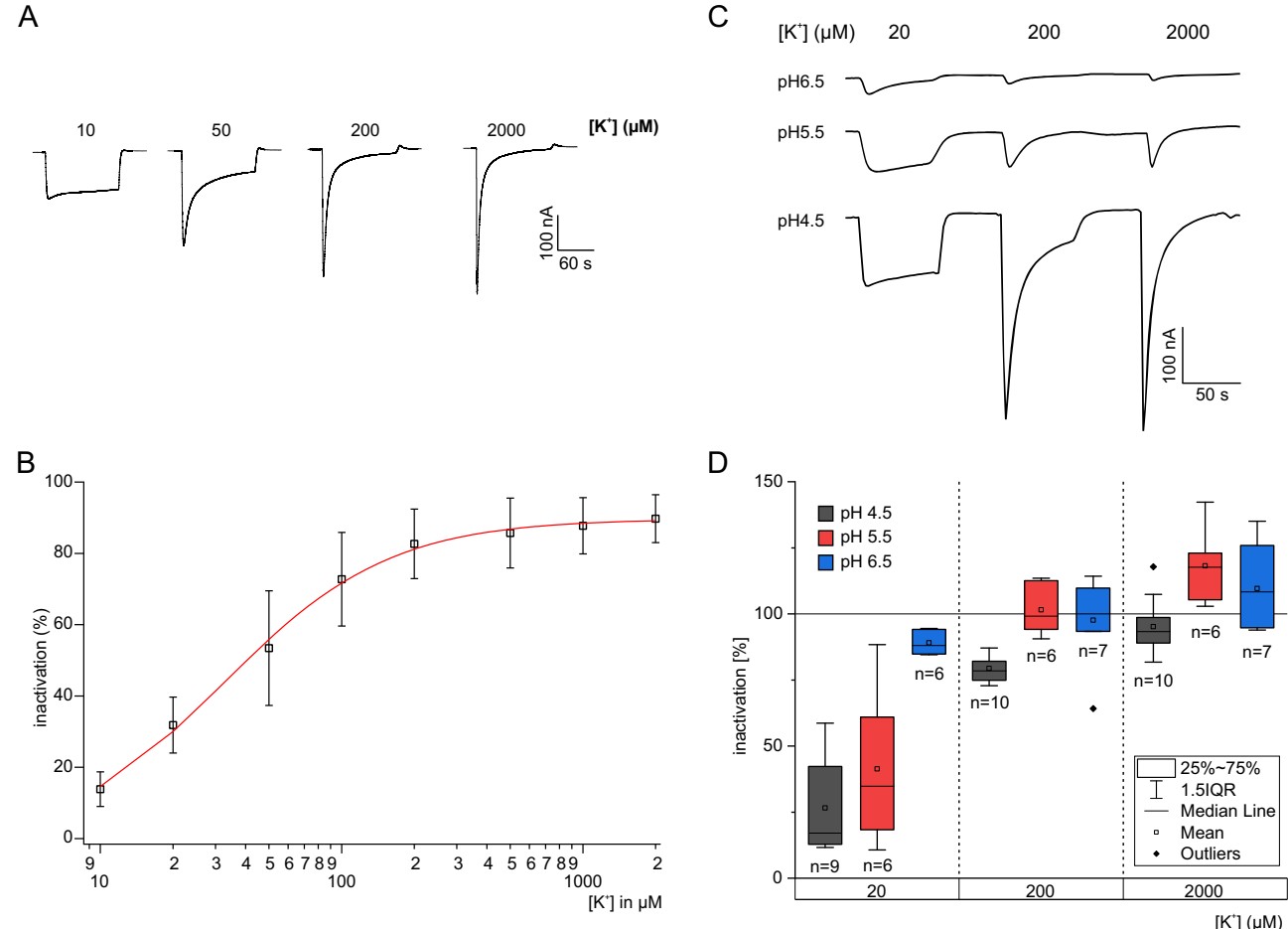

**Fig. 5 | Inactivation kinetics of AtHAK5. A** Representative current traces from oocytes co-expressing AtHAK5 and CIPK23/CBL1 at −120 mV when challenged with different K⁺ concentrations (as indicated in the figure) for 120 s. **B** Degree of inactivation (in %) derived from similar experiments as shown in A) were plotted against the applied K⁺ concentration ($n = 3$ experiments, mean ± SD). **C** Representative current response of AtHAK5/CIPK23/CBL1 co-expressing oocytes at pH4.5, pH5.5 or pH6.5 upon application of either 20, 200 or 2000 μM K⁺. **D** Box plot of the percentage of inactivation derived from similar experiments as shown in C) (number of experiments is indicated in the figure).

outward-open configuration. After binding of H⁺ (C2 state) and K⁺ (C3 state) the uncharged HAK5 transporter conducts a voltage-independent conformational change (C4 state). With the release of both ions into the cytosol the transporter has at least two negative charges and transitions back into the ground state through a voltage-dependent conformational change.

## AtHAK5 operates at low K⁺ concentrations

Sensing of external potassium concentrations in the rhizosphere is essential for plants to acclimate to varying food stocks of the macro nutrient. Thus, we asked whether AtHAK5 can adapt its activity to sudden changes in the external K⁺ concentration. To answer this question, we clamped the oocyte at −120 mV and prolonged the K⁺ incubation time to 120 s for K⁺ concentrations between 10 μM (low) and 2 mM (high). In the presence of 2 mM K⁺, AtHAK5-mediated currents reached their peak amplitude after about 2 s but vanished completely within 120 s (Fig. 5A). This response is reminiscent of K⁺-dependent inactivation of animal Shaker K⁺ channels[33,34]. When perfusing intermediate K⁺ (200 μM), a similar behavior was observed, although less pronounced in terms of degree of inactivation (Fig. 5A, B). In 10 μM K⁺, however, inward current increased, reached a steady-state amplitude, and remained constant over time (Fig. 5A, B). We named the currents that remained after the inactivation steady state currents (I_SS). Interestingly, the steady state currents plotted as a

function of the external [K⁺] resulted in a bell-shaped curve that could be well described with a modified Michaelis-Menten equation (Fig. 1E; for details see Method section). The K⁺ concentration at which the bell-shaped current reaches its maximum (28 μM) coincides with K_m [K⁺] (24 μM; Fig. 1E) supporting a correlation between the potassium-dependent increase of peak currents and the inactivation process.

When low (20 μM), intermediate (200 μM) or high (2 mM) K⁺ was applied at different pH conditions the degree of inactivation increased with the drop in external H⁺ concentrations (Fig. 5C, D). While no inactivation was observed in 20 μM K⁺ at pH4.5, currents vanished almost completely at pH6.5 at the same K⁺ concentration. Together with the finding that K⁺-induced inactivation was not terminated when the oocytes were treated with acetate (Fig. 3D), these results document that neither the cytoplasmic pH nor the PMF drive inactivation.

## Tyrosine 450 in AtHAK5 is key for K⁺ sensing

Recently, the structure of KimA, a high-affinity potassium importer from *Bacillus subtilis* was resolved by cryo-EM[27]. To obtain a mechanistic model for proton-coupled potassium symport we generated a 3D homology model for AtHAK5 using the KimA cryo-EM structure as template (Fig. 6A). Due to the low sequence homology for the cytoplasmic C-terminal domain of KimA and AtHAK5, only the transmembrane part comprising residues Q64 (equivalent to K27 in KimA) to R541 (equivalent to D465 of KimA) was modeled. The cryo-EM

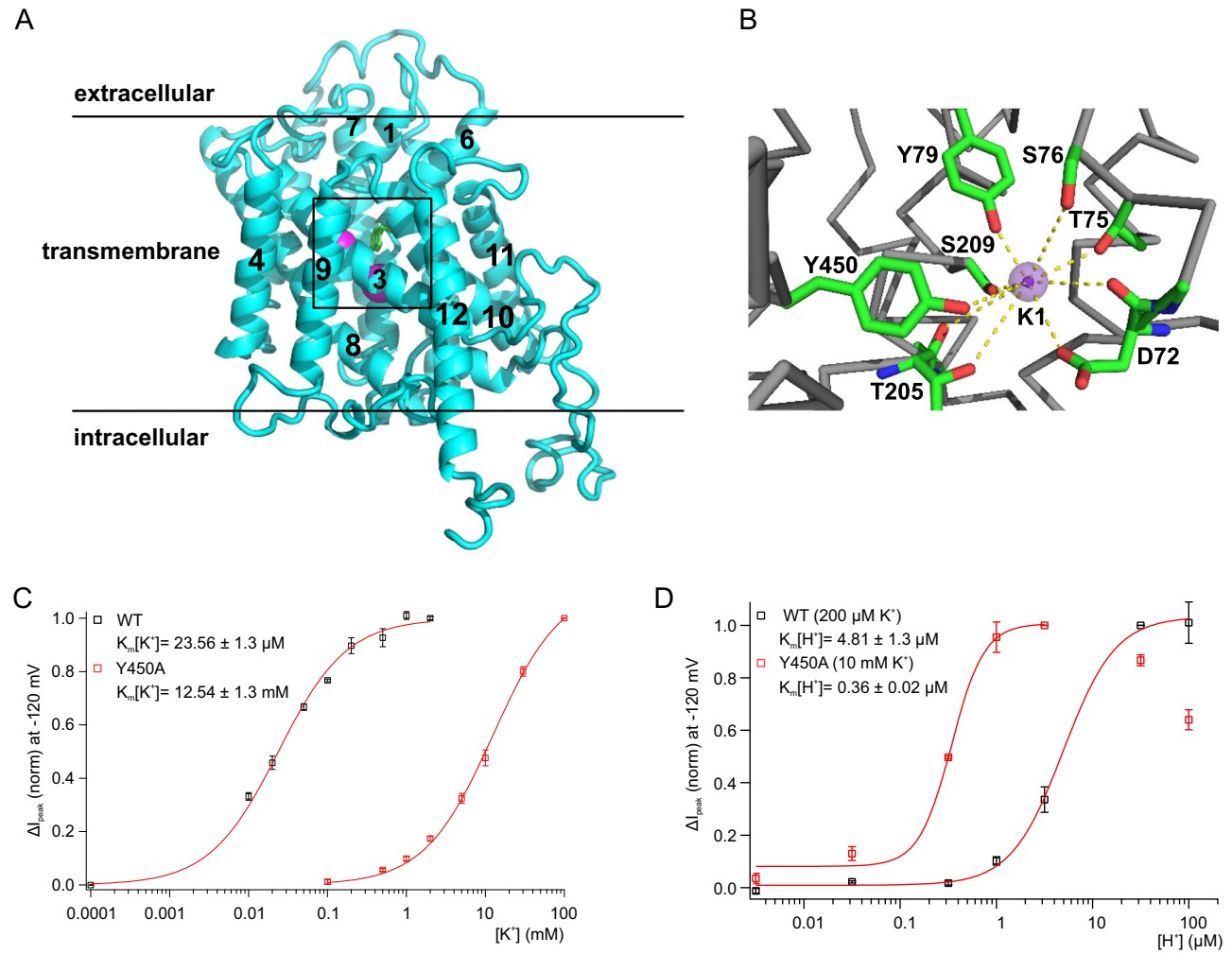

**Fig. 6 | Molecular nature of K$^+$ sensing. A** Homology model of a single subunit of AtHAK5 based on the cryo-EM structure of KimA (PDB entry 6S3K). Only the integral transmembrane part comprising residues Q64 to R541 was modeled. The helices identifiable in the chosen orientation are labeled, the indicated box shows the region magnified and shown with more details in panel **B**. The two magenta spheres represent the two potassium ions located in the upper and lower coordination site, residue Y450 is indicated in stick representation with carbon atoms colored in green. **B** Magnification of the upper ion binding site around potassium ion K1. The stringent coordination of potassium ion K1 likely facilitates dehydration of the incoming cation. **C** Normalized whole-oocyte K$^+$-induced peak currents ($\Delta I_{peak}$) at −120 mV and pH4.5 plotted against the applied K$^+$-concentration.

Currents from oocytes expressing the mutant Y450A with CIPK23/CBL1 (red squares) are compared with WT HAK5/CIPK23/CBL1 expressing oocytes (black squares, cf. Fig. 1E). K$_m$ (K$^+$) was calculated by fitting $\Delta I_{peak}$ with a Michaelis-Menten equation. ($n = 5$ experiments, mean ± SD). **D** Currents from oocytes expressing the mutant Y450A with CIPK23/CBL1 (red squares) are compared with WT HAK5/CIPK23/CBL1 expressing oocytes (black squares, cf. Fig. 3C). Normalized whole-oocyte peak currents ($\Delta I_{peak}$) were induced either by 10 mM (Y450A) or 200 μM (WT) K$^+$ at −120 mV at different pH values and plotted against the applied H$^+$-concentration. K$_m$ (H$^+$) was calculated by fitting $\Delta I_{peak}$ with a Hill equation (Hill coefficient = 2.17 ± 0.5 (Y450A) and 1.67 ± 0.3 (WT)) ($n = 5$ experiments, mean ± SD).

structure of KimA revealed three potassium ions bound inside the transmembrane part, one ion located more proximal to the extracellular side and well-coordinated by KimA residues D36, S39 (requires rotation of side chain around χ1), S40, Y43, T121, S125 and Y377. The other two potassium ions were located closer to the intracellular channel opening and are within a distance less than the sum of their two van der Waals radii, indicating that only one potassium ion is occupying this coordination site at a time[27]. While this lower cavity is wider and provides more space to the potassium ion, lower cation(s) is(are) less well coordinated than the potassium ion close to the extracellular opening. Both potassium ions are mainly coordinated by the carboxylate groups of D36 and D117 (shortest distance between one K$^+$ ion and the nearest aspartate 2.6 Å the other distance is 4 and 4.7 Å). Other polar groups, e.g., T121 or Y118 are more than 4.5 Å away. However, water molecules not modeled in the cryo-EM structure very likely occupy the cavity, in particular, if only one of the two potassium

ions is present thereby stabilizing the potassium ion coordination. Analysis of our 3D model of *Arabidopsis thaliana* HAK5 showed that potassium ion coordination is almost identical with KimA. The potassium ion proximal to the extracellular pore opening is coordinated by HAK5 residues D72, T75, S76, Y79, T205, S209 and Y450 (Fig. 6B and Supplementary Fig. 5A). The potassium ion(s) in the cavity more proximal to the cytoplasmic channel opening is(are) coordinated by D72 and D201, additional cation coordination might be from T205 (side chain hydroxyl group) and T309 (main chain carbonyl group) of HAK5 (Supplementary Fig. 5A). The upper ion binding site constitutes very likely the gate given the strong ion coordination geometry and is very likely required to achieve ion selectivity via dehydration of the incoming cation. The lower site possibly functions as electrostatic trap to facilitate ion flux direction. This site might also enable for rehydration of the cations on the transport passage towards the cytoplasm given the larger cavity size. When tested in the oocyte system

(Supplementary Fig. 5B) the exchange of D72, D201 or E312 by Ala turned AtHAK5 into an electrically silent transporter just like in KimA (Supplementary Fig. 5C, c.f.[27,35],).

In contrast to the non-functional mutations, replacement of tyrosine residue Y450 in AtHAK5 (equivalent to Y377 in KimA) by alanine did not impair transport function but suppressed the activity in 2 mM K$^+$ pH4.5 by about 90% of the WT (Supplementary Fig. 5C) and did not tend to inactivate (Supplementary Fig. 5D). Stepwise increasing the external K$^+$ concentration to 5, 10, 30 and 100 mM, the HAK5 Y450A current at pH4.5 increased in a Michaelis-Menten fashion characterized by a $K_m$ of 12.54 ± 1.3 mM (Fig. 6C). The shift towards an about 500-fold lower K$^+$ affinity was accompanied by a pronounced inactivation at 100 mM K$^+$ that increased in amplitude during the steps to 30 and finally 100 mM (Supplementary Fig. 5D and E). Plotting the steady state currents as a function of the external [K$^+$] resulted in a bell-shaped curve reaching its maximum at a K$^+$ concentration that coincides with $K_m$ [K$^+$] (12.5 mM) (Supplementary Fig. 5F) reminiscent of the WT (cf. Figure 1E). This documents that i) the conserved K$^+$ sensing sites in KimA and HAK5 allow for high affinity K$^+$ transport and ii) the same site is controlling K$^+$ sensing and inactivation in HAK5.

When comparing the current response of the mutant Y450A in the presence of different cations, K$^+$ and Rb$^+$ triggered similar current amplitudes (Supplementary Fig. 6A). However, in contrast to the WT, Cs$^+$ and NH$_4^+$ elicited only 22.6 ± 3.8 % and 20.1 ± 4.18 % of K$^+$ induced currents (cf. Fig. 2A). Similar to K$^+$ (Fig. 6C) a strong reduction in binding affinity of Y450A to all other transported cations was calculated. Whereas $K_m$ [Rb$^+$] (22.8 ± 1.3 mM) was in the same concentration range as $K_m$ [K$^+$] (Supplementary Fig. 6B, cf. Figure 6C) the binding affinities for NH$_4^+$ (309.6 ± 88.2 mM) and Cs$^+$ (78.6 ± 10.8 mM) were shifted to even higher concentrations, explaining the strong reduction in current amplitudes at 100 mM compared to K$^+$ (Supplementary Fig. 6C, D). In line with our findings for WT HAK5, inactivation of the mutant Y450A could be observed at cation concentrations higher than $K_m$, therefore, NH$_4^+$ and Cs$^+$ mediated currents did not inactivate in the tested concentration range (Supplementary Fig. 5D, E, Supplementary Fig. 6B–D).

In contrast to K$^+$, the Y450A mutant was characterized by a strong increase in H$^+$ binding affinity (Fig. 6D). Compared to the WT, the $K_m$ [H$^+$] shifted by one pH unit from 4.81 ± 1.3 μM to 0.36 ± 0.02 μM H$^+$. This strong shift in K$^+$ and H$^+$ affinities of the HAK5 Y450A mutant compared to the WT transporter could be well simulated with our four-state kinetic model by varying only a single rate constant, i.e. the binding constant of external potassium (see Supplementary Table 1 and Methods section). In agreement with the experimental data (Fig. 6C, D) the simulated currents at −120 mV with varying K$^+$ concentrations show that the decrease in K$^+$ binding affinity feeds back on the binding of protons, by increasing the apparent affinity constant of external protons (Supplementary Fig. 7A, B; for details see Methods section).

This interrelation between proton and potassium binding could be demonstrated by analysis of K$^+$ dose-response curves at different external pH values. In WT HAK5 the decrease of $K_m$ [K$^+$] from 23.56 ± 1.3 μM at pH4.5 (Figs. 1E) to 13.73 ± 3.54 μM K$^+$ at pH5.5 (Supplementary Fig. 7C, D) documents that the H$^+$ concentration feeds back on the AtHAK5 K$^+$ binding affinity.

The external pH was also found to regulate the K$^+$ affinity of the Y450A mutant as $K_m$ [K$^+$] decreased from 12.54 ± 1.3 mM at pH4.5 (Fig. 6C) to 6.19 ± 2.1 mM K$^+$ at pH5.5 (Supplementary Fig. 7E, F). When stepping the pH from 5.5 to 4.5 and finally 4.0 the decrease in K$^+$ affinity was accompanied by a reduction in K$^+$ currents (Fig. 6D). At the latter pH K$^+$ currents mediated by the Y450A mutant also lacked inactivation (Supplementary Fig. 7G). This behavior indicates that position 450 is central to K$^+$ sensing as well as to coordinate K$^+$ and H$^+$ coupling.

## Discussion

To properly study the regulation and transport properties of the prototypical HAK5-type transporter, K$^+$ electrical current recordings are indispensable[10,17]. To accomplish this task, we based our study on two-electrode voltage clamp (TEVC) analysis of AtHAK5 in Xenopus frog oocytes, a plant background-free expression system. This way, we were able to resolve ion currents mediated by AtHAK5 giving rise to a CIPK23/CBL1 activated K$^+$/H$^+$-symporter module (Fig. 1B[11],). The calculated affinity for K$^+$ uptake of 24 μM (pH4.5) (Fig. 1E) identifies AtHAK5 as high-affinity transporter (Supplementary Fig. 7D)[4].

In contrast to the AKT1 K$^+$ channels and AMT1-type NH$_4^+$ channels, AtHAK5 transport, in addition to voltage, is driven by the PMF. Based on thermodynamic considerations AtHAK5 allows uphill transport against larger K$^+$ gradients as AKT1. With a cytosolic potassium concentration of 150 mM, considering a membrane voltage of −180 mV, AKT1 would enable potassium uptake for external K$^+$ concentrations equal to or greater than 150 μM. HAK5, with its ability to couple a proton to a potassium ion, utilizing a 2-unit difference in pH between the cytosolic and the external solutions, can theoretically accumulate potassium at a concentration one hundred thousand times smaller, namely equal to or greater than 1.5 nM.

When mimicking Arabidopsis in vivo whole roots K$^+$ (Rb$^+$) uptake experiments in oocytes expressing AtHAK5[15,26], the addition of K$^+$ at any concentration did not increase currents mediated by the high-affinity K$^+$ transporter in the presence of 1 mM NH$_4^+$ (Supplementary Fig. 2E). In other words, high NH$_4^+$ concentrations outcompete access to binding sites in the permeation path of AtHAK5 for substrates other than NH$_4^+$.

The kinetic model that perfectly describes the connection between K$^+$ and H$^+$ binding/transport and the membrane potential (Supplementary Fig. 4A-D) predicts that in the outward-open configuration negatively charged amino acids (Asp72, Glu312) are accessible for binding of extracellular K$^+$ and H$^+$ (C1 state). After binding of H$^+$ (C2 state) and K$^+$ (C3 state) the uncharged transporter executes a voltage-independent conformational change into the inward-open configuration to release K$^+$ and H$^+$ into the cytosol (C4 state). The resulting conformational change from the re-charged transporter in the C4 state back into C1, is the voltage-dependent step of the transport cycle. For the homologous HAK5 transporter KimA from *Bacillus subtilis* the charged amino acids Asp36 and Glu233 were found important for binding of K$^+$ and/or H$^+$[27]. The single exchange of the respective residues in AtHAK5 (Asp72, Asp201 and Glu312; see Fig. 6A, B, Supplementary Fig. 5A) by Ala resulted in unfunctional AtHAK5 mutants (Supplementary Fig. 5C, c.f[35].) supporting that our kinetic model is correct. In the KimA structure, Tyr377 was identified to coordinate the binding of K$^+$[27]. Our results show that the respective Tyr450 in AtHAK5 (Fig. 6A, B, Supplementary Fig. 5A) is crucial for sensing micromolar K$^+$ concentrations as the binding affinity for K$^+$ of the mutant Y450A strongly decreased by factor 500 shifting into the millimolar range whereas the affinity to bind H$^+$ increased (Fig. 6C, D).

The exceptional feature of AtHAK5 to inactivate at high K$^+$ concentrations (Fig. 1E, Fig. 5) and to activate at low K$^+$ concentrations (Fig. 1E, Supplementary Fig. 1) combines the K$^+$ transport function and K$^+$ sensing ability within one protein. As soon as the K$^+$ concentration drops to the low micromolar range, K$^+$ transport via AKT1 ceases (Fig. 1A). While AKT1-mediated K$^+$ transport vanishes, the drop in external K$^+$ leads to the activation of AtHAK5 (Fig. 1D, Supplementary Fig. 1) maintaining sufficient K$^+$ uptake even at low μM K$^+$ concentrations. Under raising concentrations of external K$^+$, HAK5 inactivates thus saving energy by keeping the proton gradient unaffected and preventing excessive membrane depolarization and cytosolic acidification under conditions that allow K$^+$ uptake via the hyperpolarization-activated shaker channel AKT1. Otherwise, the HAK5 mediated depolarization of the plasma membrane would counteract AKT1 activity until HAK5 is eventually inactivated by a

protein phosphatase or degraded, similar to the transceptor BOR1[36] that promotes own ubiquitination and degradation according to local B concentrations. Thus, B-sensing by the transceptor BOR1 is coupled with a feedback loop resulting in a reduction in B-transport. Transceptors are membrane proteins that carry out both transport and signaling functions[37] as it was shown for human and yeast amino acid transporters SLC38A9[38] or Gap1[39,40], respectively. Transceptor mediated downstream signaling is often linked to transport-dependent conformational changes, e.g. structural transition and protomer coupling of nitrate transceptor NRT1.1 are promoted by nitrate binding and were found essential for NRT1.1 function as sensor[41,42].

Just recently, Dreyer et al.[43] could demonstrate that a $K^+$-homeostat (the sum of $K^+$ channels and transporters), in combination with the proton pump, could exhibit transceptor-like characteristics. The signal of a relative change in external $K^+$ can be converted into a change in cytosolic pH. Because regulation of cytosolic pH might be coupled to cellular metabolism[44] or anion channel activity[31], the change in extracellular $K^+$ sensed by the $K^+$-homeostat might thus directly influence cellular processes. As AtHAK5 is the main player under $K^+$ starvation, mediating $K^+/H^+$ symport and transport-mediated cytosolic pH changes (Fig. 3A), AtHAK5 could be key in such a transceptor network. Feedback regulation of this $K^+$ homeostat most probably also involves $Ca^{2+}$ signaling as $K^+$ uptake via HAK5 and AKT1 is under direct control of the CIPK/CBL network[11,12]. The activity of this $Ca^{2+}$ signaling hub depends on the plant $K^+$ status and is tightly regulated by a set of phosphatases[45] and the TOR complex[46], shown to connect energy and nutrient status to cell growth.

However, the direct link between HAK5 mediated $K^+$ sensing, regulation of transport activity and putative downstream signaling is still scant. Thus, the question remains whether these properties indicate that HAK5 is not only $K^+$-selective transporter, but also transceptor. An exciting venue of future research will be to analyze the transcriptome of *hak5* loss-of-function mutant plants to determine to what extend the transcriptional response to $K^+$-starvation is missing in the mutant, similar to what has been determined for the nitrate transporter and sensor NRT1.1/NPF6.3[47,48].

The biophysical information about the function of AtHAK5 activating under soil $K^+$ starvation provides breeders with valuable knowledge to improve plant low $K^+$ resilience and to reduce the use of $K^+$ fertilizers.

## Methods

### Experimental model and subject details
TEVC experiments utilized oocytes from healthy, non-immunized female adult Xenopus laevis frogs at the Julius-von-Sachs Institute, Wuerzburg University. Permission to keep and use these animals for partial ovariectomy is registered at the government of Lower Franconia, ref. No. 55.2.2-2532-2-1850-11 and were performed following the guidelines of the European animal welfare law. Mature female *X. laevis* frogs (male Xenopus frogs do not produce oocytes) were kept at 20 °C at a 12/12 h day/night cycle in dark grey 96 liters tanks (5 frogs/tank). Frogs were fed twice a week with floating trout food (Fisch-Fit Mast 45/7 2 mm, Interquell GmbH, Wehringen, Germany). Tanks are equipped with 30 cm long PVC pipes with a diameter of around 10 cm used as hiding places for the frogs. The water is continuously circulated and filtered by a small aquarium pump.

### Xenopus oocyte preparation
For oocyte isolation, mature female *X. laevis* frogs were anesthetized by immersion in water containing 0.1% 3-aminobenzoic acid ethyl ester. Following partial ovariectomy, oocytes were treated with 0.14 mg/ml collagenase I in $Ca^{2+}$-free ND96 buffer (10 mM HEPES pH 7.4, 96 mM NaCl, 2 mM KCl, 1 mM $MgCl_2$,) for 1.5 h. Subsequently, oocytes were washed with $Ca^{2+}$-free ND96 buffer and stage V or VI oocytes were kept at 16 °C in ND96 solution (10 mM HEPES pH 7.4,

96 mM NaCl, 2 mM KCl, 1 mM $MgCl_2$, 1 mM $CaCl_2$) containing 50 mg/l gentamycin. For electrophysiological experiments, 10 ng of each cRNA was injected into selected oocytes. Oocytes were incubated for 2 to 4 days at 16 °C in ND96 solution.

### Methods details
**Cloning, site-directed mutagenesis and cRNA synthesis.** The complementary DNA (cDNA) of AtHAK5 WT was cloned into oocyte expression vectors (pNB1, based on pGEM vectors), by an advanced uracil-excision-based cloning technique as described by ref. 49. Site-directed mutations were introduced by means of a modified USER fusion method as described by refs. 50,51. Primers used for cloning and site-directed mutagenesis are listed in Supplementary Table 2. In brief, the coding sequence of AtHAK5 within an oocyte expression vector (based on pNBIu vectors) was used as a template for USER mutagenesis. Overlapping primer pairs (overlap covering 8 to 14 bp including the mutagenesis site) were designed[51]. PCR conditions were essentially as described by ref. 50 using PfuX7 polymerase. PCR products were treated with the USER enzyme (New England Biolabs, Ipswich, MA, USA) to remove the uracil residues, generating single-stranded overlapping ends. Following uracil excision, recirculation of the plasmid was performed at 37 °C for 30 minutes followed by 30 minutes at room temperature, and then constructs were immediately transformed into a chemical competent *Escherichia coli* cells (XL1-Blue MRF'). All mutants were verified by sequencing.

For functional analysis in *Xenopus* oocytes, complementary RNA (cRNA) was prepared with the AmpliCap-Max T7 High Yield Message Maker Kit (Cellscript, Madison, WI, USA).

### Oocyte assays
**Two-electrode voltage-clamp (TEVC) studies.** Oocytes were perfused with Mes/Tris-based buffers. For HAK5 WT measurements, the standard solution contained 10 mM Mes/Tris (pH 4.5), 1 mM $CaCl_2$, 1 mM $MgCl_2$, 0.1 mM $LaCl_3$ 220 mM Sorbitol and 2 mM LiCl, KCl, NaCl, RbCl, CsCl or $NH_4Cl$. To balance the ionic strength, we compensated for changes in the cation concentration with LiCl. For characterization of the HAK5 Y450A mutant, standard solutions contained 10 mM Mes/Tris (pH 4.5), 1 mM $CaCl_2$, 1 mM $MgCl_2$, 0.1 mM $LaCl_3$ and 100 mM KCl. To balance the ionic strength, we compensated for changes in the cation concentration with LiCl.

Standard measurements: Oocytes were clamped to −120 mV and current response was recorded in the presence of different cations, different cation concentrations and/or different pH-values.

Determination of cation selectivity: AtHAK5 WT: Whole-oocyte currents in the presence of different cations (2 mM) of either water-injected control oocytes or oocytes co-expressing AtHAK5 and CIPK23/CBL1 were recorded. Average current response of control oocytes ($Li^+$: −94.83 ± 9.39 nA, $Na^+$: −95.43 ± 7.71 nA, $K^+$: −84.55 ± 7.2 nA, $Rb^+$: −81.66 ± 6.15 nA, $Cs^+$: −92.19 ± 5.55 nA, $NH_4^+$: −65.12 ± 2.95 nA; $n = 3$ experiments, mean ± SD) was subtracted from AtHAK5 mediated currents. Mutant Y450A: Whole-oocyte currents in the presence of different cations (100 mM) of either water-injected control oocytes or oocytes co-expressing AtHAK5 Y450A and CIPK23/CBL1 were recorded. Average current response of control oocytes ($Li^+$: −58.08 ± 5.6 nA, $Na^+$: −72.91 ± 5.6 nA, $K^+$: −73.28 ± 10.1 nA, $Rb^+$: −77.03 ± 9.45 nA, $Cs^+$: −79.11 ± 17.7 nA, $NH_4^+$: −72.51 ± 14.9 nA; $n = 4$ experiments, mean ± SD) was subtracted from AtHAK5 Y450A mediated currents.

Determination of steady-state currents ($I_{SS}$): Starting from a holding potential of 0 mV in the presence of different $K^+$ concentrations, a voltage jump to −120 mV was applied for 120 s before jumping back to the holding potential. $I_{SS}$ was extracted at the end of the −120 mV pulse.

To calculate the pH-dependent half-maximal AtHAK5 activity ($K_m$ ($H^+$)), data points were fitted with a Hill equation. To calculate the cation-dependent half-maximal AtHAK5 activity ($K_m$ [$K^+$]/[$NH_4^+$]/[$Rb^+$]/[$Cs^+$]),

HAK5-mediated peak currents were fitted by the Michealis-Menten equation:

$$\Delta I_{peak} = I_{max\_norm} \frac{[K^+]}{[K^+] + K_m} \quad (1)$$

obtaining $I_{max\_norm} = 1.01 \pm 0.01$ $K_m = 28.3 \pm 0.9$ mM for the WT and $I_{max\_norm} = 1.12 \pm 0.03$ $K_m = 12.92 \pm 0.5$ mM for the mutant Y450A. The latter value was held constant for fitting the steady-state data, which was performed using the following equation:

$$I_{ss} = \frac{[K^+]}{[K^+] + K_m} \frac{I_{max}}{\frac{[K^+]}{K_m} + 1} + I_s \quad (2)$$

We obtained $I_{max} = 1.01 \pm 0.16$ µA and $I_s = 124 \pm 17$ nA for the WT and $I_{max} = 1.00 \pm 0.01$ µA and $I_s = 0$ nA for the mutant Y450A. The concentration at which the bell-shaped current reaches its maximum coincides with $K_m$ supporting a correlation between the potassium-dependent increase of peak currents and the inactivation process.

Determination of percentage of inactivation: Percentage of inactivation was calculated via ($\Delta I_{peak} - \Delta I_{SS}$)/$\Delta I_{peak}$ multiplied by 100.

**Extra- and intracellular $K^+$ and $H^+$ measurements during TEVC measurements.** Electrodes were pulled from borosilicate glass capillaries (KWIK-FIL TW F120-3 with filament) with a vertical puller (Narishige Scientific Instrument Lab), baked for a minimum 2 hr at 220 °C and silanized for 1 hr with dimethyldichlorosilane (Silanization Solution I, Sigma Aldrich). For $K^+_{ext}$ measurements, $K^+$ selective electrodes were backfilled with a buffer containing 10 mM KCl and 5 mM CaCl$_2$. The electrode tip was filled with a $K^+$-selective ionophor cocktail (potassium ionophore I cocktail B, Sigma-Aldrich) by dipping the tip into the cocktail. Electrodes were calibrated in standard buffers containing 20 µM, 200 µM, 2 mM or 20 mM KCl at pH4.5 before and after each measurement. $H^+$ selective electrodes were backfilled with a buffer containing 40 mM KH$_2$PO$_4$, 23 mM NaOH, and 150 mM NaCl (pH 7.5). The electrode tip was filled with a $H^+$ selective ionophor cocktail (hydrogen ionophore I cocktail A, Sigma-Aldrich) by dipping the tip into the cocktail. Electrodes were calibrated in 2 mM KCl at pH 5.5, 6.5, and 7.5 before and after each measurement. pHluorin-based recordings of pH$_{cyt}$: pHluorin2 (pH2) was fused N-terminally to HAK5 as described previously[31]. Oocytes expressing the pH2-HAK5 construct were illuminated with 400 and 470 nm and emission from 510 to 550 nm was collected with a CCD camera. A change in cytosolic pH was represented by a change in the fluorescence ratio between 400 nm/470 nm.

**Homology modeling of Arabidopsis thaliana HAK5.** A 3D homology model of HAK5 from Arabidopsis thaliana comprising residues Q64 to R541 was done manually. First a multiple sequence alignment employing ClustalOmega was done using the amino acid sequences from HAK5 from Arabidopsis thaliana, HAK5, HAK16, HAK22, HAK27 from Oryza sativa, KUP12 from Arabidopsis thaliana, and KIM sequences from Bacillus subtilis, Bacillus licheniformis, Bacillus halotolerans, Bacillus atrophaeus and Bacillus velezensis. Since the N-terminal and C-terminal sequences of HAK and KimA channel proteins differed significantly, the amino acid sequences for subsequent alignments were trimmed to cover residues L61 to R631 for HAK5 and L27 to L539 of KimA, the latter of which presents the structured part as found in the cryo-EM structure of Kim (PDB entry code 6S3K, Tascon et al., 2020). The final multiple sequence alignment obtained with the trimmed sequences for HAK5 was then used to replace the amino acid of KimA –the structure of KimA (PDB entry 6S3K) was used as template– with the respective amino acid residue of HAK5 as found in the sequence alignment. The obtained 3D homology model was then refined stepwise using the software Quanta version 2005 and the CHARMM module. The CHARMM27 force field was used, first all heavy atoms were kept fixed and only hydrogen atoms were minimized using short energy minimization (200 steps of Adopted Raphson Newton algorithm). In subsequent energy minimization (each time 200 steps of Adopted Raphson Newton minimization), side chain heavy atoms were first restrained by a harmonic potential of 25 kcal mol$^{-1}$ Å$^{-2}$, which was lowered to 10 and then to 0 kcal mol$^{-1}$ Å$^{-2}$ to minimize drift from the original coordinates. Only geometrical energy terms with a van der Waals cutoff of 11 Å without energy terms for electrostatic interactions were used. The final 3D model of Arabidopsis thaliana HAK5 exhibited reasonable backbone and sidechain geometry. While the initial model also contained parts of the C-terminal cytoplasmic domain, which is in the KimA channel protein is involved in dimer formation, the model was shortened and comprises the residues Q64 to R451. The model was used as monomer due to the absence of the cytoplasmic domain; the three potassium ions present in the KimA template were also modeled in the HAK5 homology model since the residues surrounding the $K^+$ ions are highly conserved in HAK5. As the two potassium ions occupying the coordination site closer to the cytoplasmic pore opening share a distance to each other less than the sum of their van der Waals radii (see also Tascon et al., 2020), only one potassium ion was placed in the lower coordination site. Hence, the final 3D homology model of HAK5 comprises residues Q64 to R451 and two potassium ions, of which one occupies the coordination site close to Y450 and the other potassium ion is located in the lower coordination site close to D201.

**Kinetic model for HAK5-mediated peak currents.** To describe our experimental data, particularly the different voltage dependence of potassium and proton currents shown in Fig. 4A-E, we start with the six-state kinetic scheme depicted in Supplementary Fig. 4A, which has been proposed in algae, yeast, and plant cells (see[52] and references therein). This scheme posits that external protons ($[H^+]_o$) bind to the empty transporter, facing the extracellular side (transition from C1 to C2), followed by binding with external potassium ($[K^+]_o$, transition from C2 to C3). A conformational change (transition from C3 to C4) allows the transporter to release potassium ($[K^+]_i$, transition from C4 to C5) first and then protons ($[H^+]_i$, the transition from C5 to C6) into the cytosol. A conformational change drives the empty transporter facing the cytosolic face, state C6, to C1. Since our interest lies in studying the dependence of currents on external protons and potassium, we can further simplify the model lumping C4, C5, and C6 in a single state (C4) as shown in Supplementary Fig. 4B and C. This scheme has four states; let's assume there is a single voltage-dependent step between C1 and C4 through a single symmetric Eyring barrier (see[53], for various versions of this scheme). Therefore, states C1 and C4 each have two negative charges, state C2 has one negative charge, while C3 has no net charge. The two rate constants, k14 and k41, are as follows:

$$k_{14} = k_{14}^0 e^{\frac{FV}{RT}} \text{ and } k_{41} = k_{41}^0 e^{-\frac{FV}{RT}} \quad (3)$$

while

$$k_{43} = k_{43}^0 [H^+]_i [K^+]_i \quad (4)$$

This model is consistent with the high-resolution structure of a homologous HAK5 transporter[27]: in the absence of protons and potassium, the transporter contains negatively charged amino acids (aspartate and glutamate) that can face either the extracellular side of the membrane (C1 state) or the cytosolic side (C4 state). When the transporter is in the C1 state, protons enter and bind to the innermost glutamate (C2 state). Then, potassium enters and interacts with an aspartate (C3 state). In the C3 state, the transporter has no net charge, and a voltage-independent conformational change occurs, leading to the release of protons and potassium into the cytosol. The transporter

in the C4 state has at least two negative charges and transitions to the C1 state through a voltage-dependent conformational change.

As far as our model is concerned, the principle of microscopic reversibility imposes that not all kinetic constants are independent, meaning:

$$k_{41}^0 k_{12}^0 k_{23}^0 k_{34} = k_{14}^0 k_{21} k_{32} k_{43}^0 \tag{5}$$

The current mediated by the transporter is as follows:

$$I = 2eN(k_{14}C_1 - k_{41}C_4) \tag{6}$$

where $e$ is the elementary charge and N is the total number of transporters present on the oocyte membrane. Assuming a steady-state condition, the kinetic scheme can be analytically solved using the King and Altman method[54], as shown in Supplementary Fig. 4D. The result obtained is:

$$I = 2eN k_{41}^0 k_{12}^0 k_{23}^0 k_{34} \frac{\left([H^+]_i[K^+]_i e^{\frac{FV}{RT}} - [H^+]_o[K^+]_o e^{-\frac{FV}{RT}}\right)}{\sum_{i=1}^{4}\sum_{j=1}^{4} C_{i,j}} \tag{7}$$

The sixteen terms in the denominator, $C_{i,j}$, obtained through the procedure outlined in Supplementary Fig. 4D, are as follows:

$$C_{1,1} = k_{23}^0[K^+]_o k_{34} k_{41},\ C_{1,2} = k_{34}k_{41}k_{21},\ C_{1,3} = k_{41}k_{21}k_{32},\ C_{1,4} = k_{21}k_{32}k_{43} \tag{8}$$

$$C_{2,1} = k_{34}k_{41}k_{12}^0[H^+]_o,\ C_{2,2} = k_{41}k_{12}^0[H^+]_o k_{32},\ C_{2,3} = k_{12}^0[H^+]_o k_{32} k_{43}, \\ C_{2,4} = k_{32}k_{43}k_{14} \tag{9}$$

$$C_{3,1} = k_{41}k_{12}^0[H^+]_o k_{23}^0[K^+]_o,\ C_{3,2} = k_{12}^0[H^+]_o k_{23}^0[K^+]_o k_{43}, \\ C_{3,3} = k_{23}^0[K^+]_o k_{43}k_{14},\ C_{3,4} = k_{43}k_{14}k_{21} \tag{10}$$

$$C_{4,1} = k_{12}^0[H^+]_o k_{23}^0[K^+]_o k_{34},\ C_{4,2} = k_{23}^0[K^+]_o k_{34}k_{14},\ C_{4,3} = k_{34}k_{14}k_{21}, \\ C_{4,4} = k_{14}k_{21}k_{32} \tag{11}$$

The probability that the system is in state $C_i$ (i = 1, 2, 3, and 4) is given by:

$$C_i = \frac{\sum_{j=1}^{4} C_{i,j}}{\sum_{i=1}^{4}\sum_{j=1}^{4} C_{i,j}} \tag{12}$$

where each $C_{i,j}$ is obtained from Eqs. 8–11.

Equation 7 can be written as follows:

$$I = I_+ + I_- \tag{13}$$

where

$$I_+ = 2eN k_{41}^0 k_{12}^0 k_{23}^0 k_{34} \frac{[H^+]_i[K^+]_i e^{\frac{FV}{RT}}}{\sum_{i=1}^{4}\sum_{j=1}^{4} C_{i,j}} \tag{14}$$

represents the movement of protons and potassium from the cytosol to the outside of the oocyte and is, therefore, a positive current, while

$$I_- = -2eN k_{41}^0 k_{12}^0 k_{23}^0 k_{34} \frac{[H^+]_o[K^+]_o e^{-\frac{FV}{RT}}}{\sum_{i=1}^{4}\sum_{j=1}^{4} C_{i,j}} \tag{15}$$

represents the entry of protons and potassium into the cytosol from the external solution.

Equation 7 has eight free parameters, seven rate constants (the eighth is obtained from the principle of microscopic reversibility, Eq. 5), and the number of transporters, N, expressed on the oocyte membrane.

**Data Simulation.** If the free parameters of Eq. 7 are assigned the values shown in the left column of Supplementary Table 1, and Eq. 10 is used to simulate the data in Fig. 4A and D, the simulation results are represented by the dashed lines in Supplementary Fig. 4E and F. The solid lines are obtained from Eq. 18 and are either overlapped or very close. Equation 15 can be rewritten as follows:

$$I = I_{X\max} \frac{[X^+]_o}{[X^+]_o + K_X} \tag{16}$$

Equation 16 is a Michaelis-Menten equation, where X represents either protons or potassium. $I_{X\max}$ and $K_X$ are the maximum current and the apparent affinity for the respective ion. We can explicitly write $I_{K\max}$ and $K_K$ as follows:

$$I_{K\max} = -\frac{2eN k_{41}^0 k_{12}^0 k_{23}^0 k_{34}[H^+]_o e^{-\frac{FV}{RT}}}{k_{23}^0\left(k_{34}k_{41} + k_{14}(k_{43}+k_{34}) + k_{12}^0[H^+]_o(k_{41}+k_{43}+k_{34})\right)} \tag{17}$$

$$K_K = \frac{k_{32}\left(k_{41}k_{21}(1+\frac{k_{34}}{k_{32}}) + k_{43}k_{21} + k_{14}(k_{43}+k_{21}(1+\frac{k_{43}+k_{34}}{k_{32}})) + k_{12}^0[H^+]_o(k_{41}(1+\frac{k_{34}}{k_{32}})+k_{43})\right)}{k_{23}^0\left(k_{34}k_{41} + k_{14}(k_{43}+k_{34}) + k_{12}^0[H^+]_o(k_{41}+k_{43}+k_{34})\right)} \tag{18}$$

The simulations of Eqs. 17 and 18 are shown in Supplementary Figs. 4G and H, and they are in excellent agreement with the experimental data in Fig. 4B and C.

We can also write the equation for $K_H$ as follows:

$$K_H = \frac{k_{21}\left(k_{41}(k_{34}+k_{32}) + k_{43}k_{32} + k_{14}(k_{43}(1+\frac{k_{32}}{k_{21}})+k_{34}+k_{32}) + k_{23}^0[K^+]_o\frac{k_{41}(k_{34}+k_{43})+k_{34}k_{14}}{k_{21}}\right)}{k_{12}^0\left(k_{41}(k_{34}+k_{32}) + k_{32}k_{43} + k_{23}^0[K^+]_o(k_{41}+k_{43}+k_{34})\right)} \tag{19}$$

Supplementary Fig. 4I shows the simulation of Eq. 19 in agreement with the data from Fig. 4E.

**The mutant Y450A.** The Y450A mutation alters the interaction between external potassium and its binding site within the transporter. In the model, a similar effect can be achieved by decreasing the binding constant of external potassium, $k_{23}^0$, by a thousand times (see Supplementary Table 1). In Supplementary Fig. 7A, a comparison of the currents at −120 mV simulated with varying external potassium in the wild-type (WT) channel and the mutant is shown, in perfect agreement with the experimental data from Fig. 6C. The decrease in $k_{23}^0$ also affects the apparent affinity constant of external protons, which tends to increase, as shown in the simulations of Supplementary Fig. 7B and experimentally verified in Fig. 6D. It is worth noting that we would have obtained identical simulations by increasing the potassium debinding constant, $k_{32}$, by a thousand times.

**Statistics and reproducibility.** The current measurements of several individual oocytes have been shown to be highly reproducible. Non-injected control oocytes did not respond to any test buffer under the chosen conditions. Thus, in oocyte measurements the sample size (N) was typically chosen between 3 and 10 individual oocytes originating from at least two independent surgeries/frogs. Different oocytes from different frogs were used for randomization. Data were only excluded when for a technical reason the measurement failed. If e.g. any experimental artefacts (voltage clamping problems, substantial seal damage) within a test series were detected, the oocyte and thus the whole test series was handled as outlier and not

included in the final analysis. As a standard control, the background/leak current was measured at the beginning and end of each test series.

No statistical method was used to predetermine the sample size. The Investigators were not blinded to allocation during experiments and outcome assessment.

**Data collection and analysis.** Electrophysiological data acquisition was performed with Patchmaster (software version 2×92; Multi Channel Systems MCS GmbH, Germany). Electrophysiological data were analyzed with the software Igor Pro 8 (waveMetrics, Inc., Lake Oswego, Oregon, USA), Excel (Microsoft Corp. Redmond, Washington, USA) and Origin (OriginPro, Version 2021b. OriginLab Corporation, Northampton, MA, USA). Thermodynamic model and figures were carried out with Igor Pro software (version 6.2, Wavemetrics, Lake Oswego, OR, USA). Multiple sequence alignment was performed with ClustalOmega. For 3D homology modeling the software Quanta (version 2005, MSI Accelrys, San Diego) and the CHARMM module (version C31b1) was used. Graphics were created with PyMOL (version 1.8.4, The PyMOL Molecular Graphics System, Schrödinger, LLC.).

### Reporting summary

Further information on research design is available in the Nature Portfolio Reporting Summary linked to this article.

## Data availability

Sequence data from this article can be found in the GenBank/EMBL data libraries under accession numbers AF129478 (AtHAK5), NM_102766 (AtCIPK23), and AF076251 (AtCBL1). Correspondence and requests for materials should be addressed to the corresponding authors. The processed data that support the findings of this study are provided in the Supplementary Information/Source Data file. Source data are provided with this paper.

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

## Acknowledgements

This work was funded by the German Research Foundation (DFG) grants for the priority program 'MAdLand – Molecular Adaptation to Land: Plant Evolution to Change' HE 1640/45-1 to R.H., the DFG Reinhart Koselleck Grant 415282803 to R.H, as well as by "Progetti di Ricerca di Interesse Nazionale" grant 20222CS2B3 to A.C.

## Author contributions

R.H., D.G., J.M.P., I.H., T.D.M., A.C. and T.M. designed the study and wrote the manuscript. T.M., S.S. and A.C. performed the experiments and/or analyzed the data. A.C. established the thermodynamic model.

## Funding

## Competing interests

The authors declare no competing interests.
