## [Peer Review File · Nature Communications]

Reviewers' comments:

Reviewer #1 (Remarks to the Author):

In this very interesting manuscript, the authors show that HAK5 from the plant *A. thaliana* is a hyperpolarization activated K⁺/H⁺ symporter, coupling the inward movement of potassium against its electrochemical gradient to the inwardly directed proton motive force (PMF). A detailed characterization of the transport mechanism that includes electrophysiology, fluorescence measurements, mutagenesis, as well as kinetic and structural modeling, reveals that the transporter is able to bind K⁺ with high affinity but also responds to high K⁺ concentrations by inactivating. While dependent on K⁺ concentration, the inactivation process appears to be independent of cytoplasmic pH or PMF. Ammonium is found to be transported by AtHAK5, but not as efficiently as K⁺. Nevertheless, at high concentrations, ammonium can outcompete K⁺ for transport. Furthermore, based on homology modeling with KimA from *B. subtilis*, tyrosine 450 is found to be critical to the high affinity binding of K⁺ to AtHAK5. Overall, the manuscript is clearly written, and the conclusions are well supported by the data.

One point that needs to be addressed concerns the classification of AtHAK5 as a tranceptor. In the manuscript, the word tranceptor appears to be used as a synonym for high affinity transporter. While tranceptors do have a high affinity site for the ion or molecule they transport, they also directly or indirectly affect downstream effectors through signaling cascades like receptors. In the literature cited by the authors for the definition of tranceptor (refs. 36-37), intracellular signaling is mediated by phosphorylation in Ho et al. Cell 2009, or by second messengers, Ca²⁺ or H⁺, in Dreyer et al. iScience 2022. In their description of AtHAK5, the authors show that PMF is a driver for the movement of potassium against its electrochemical gradient. However, it is unclear whether protons serve the additional role of second messengers for AtHAK5. Are there pH-dependent cellular events that can be linked to the activity of AtHAK5 under low K⁺ conditions?

The authors should explain the K⁺/H⁺ stoichiometry they assumed in the kinetic modeling of transport. Also, it is not clear why they initially have a 6-state model in Fig. S2. Why is the value for the $k_{2,1}$ not provided in supplemental table 1?

Why was a Boltzmann function used to fit the data in Fig 3B, instead of a Hill equation? The methods section states that the Hill equation was used to calculate the pH-dependent half maximal activity of AtHAK5. If that is the case, the value of the Hill coefficient derived from the fit should be provided as well.

The Michaelis-Menten equation used to calculate I_{peak} in the methods section seems to be missing a current term. It could only be correct the way it is written now if I_{peak} were somehow normalized and rendered dimensionless.

Minor points

On page 3, akt1-1 is not introduced.

It is recommended to maintain consistency about axis scales. In Fig.2B, the scale of the x-axis is not logarithmic whereas it is logarithmic in equivalent graphs in all other figures.

When the authors write “opus operandi”, they probably mean “modus operandi.”

The word “blotted” is used in multiple places in the figure legends. It should be replaced with “plotted.”

Reviewer #2 (Remarks to the Author):

In their study, Maierhofer et al. present the biophysical properties of AtHAK5, a member of the KUP/HAK/KT family of potassium transporters, by means of heterologous expression in *Xenopus* oocytes and two-electrode voltage clamp. They found the hallmark properties of a high-affinity K⁺/H⁺ symporter. An interesting feature is the presence of an inactivation process at elevated K⁺ concentrations, at which apparently a K⁺-selective ion channel takes over in vivo to accomplish root K⁺ uptake. The molecular mechanisms of symport and inactivation were investigated using kinetic modeling, structural modeling and site-directed mutagenesis.

This is a solid biophysical study going deep into the molecular mechanisms of AtHAK5 function. The manuscript is for the most part clearly written and presented. Some points however require to be clarified or corrected.

1. Much weight is laid on the definition of AtHAK5 as a K⁺-sensing transceptor. Is this justified? It is clear that every transport protein in some way “senses” its substrate. Here, high external K⁺ concentrations trigger an intrinsic inactivation process in the same transport protein, which may make sense from a physiological and energetical point of view. However, there is no indication that this information on the K⁺ status is transmitted to the cell and influences secondary processes.

2. The Y450A mutant showed an interrelation between K⁺ and H⁺ binding, also supported by the kinetic model. This is an interesting finding, but was not explained further. How could this work mechanistically?

3. Does the Y450A exchange (Figure 6) equally affect the other cation substrates (Figures 2, S1) in terms of affinity and inactivation?

4. The pH dependence experiments (Figure 6D) was performed at 200 micromolar K⁺ for the WT (corresponding to about 10 times the K_m for K⁺) and 10 mM for the Y450A mutant (corresponding to the K_m for K⁺). Can these different conditions be compared?

5. Lines 94/95: It is stated that in oocytes co-expressing AtHAK5 and CBL1 “no transport activity” could be recorded at 2 mM K⁺. But see Figure 1B! Please clarify!

6. Order of panels in Figure 2: the authors may consider to change panel C to A, in order to comply with the order in the text.

7. It is stated that both Cs⁺ currents (line 108) and NH₄⁺ currents (line 115) were “about 40% smaller” than K⁺ currents. In Figure 2C, Cs⁺ currents and NH₄⁺ currents are not equal though. Please correct!

8. The part on the pH dependence (lines 126-150) should be written more clearly. Convincing evidence for a K⁺/H⁺ symport mechanism comes from the concomitant cytosolic acidification during inward K⁺ transport (Figure 3C). Measurements at a fixed external (Figure 3A/B) or internal pH (Figure 3D) may simply indicate pH-dependent modulation.

9. At several positions within the manuscript, please change “blotted against..” into “plotted against..”

Reviewer #3 (Remarks to the Author):

This work shows quite elegantly that the HAK5 transport mechanism is K/H symport. HAK5 activity was maximal at μM K concentrations and decreased in the mM range in agreement with the external K concentration at which this transporter operates in Arabidopsis roots. HAK5 was activated by membrane hyperpolarisation and by external acidic pH by a mechanism separate from the mere electrophoretic pull of protons. Authors document that ammonium is a physiologically relevant substrate of HAK5. Last, HAK5 transport inactivated in a K- and time-dependent manner, and residue Y450 involved in K binding was critical for this feature. Authors propose that HAK5 is a transceptor that operates as a low-K sensor.

Comments ordered in decreasing order of importance:

1. Authors conclude that HAK5 is a K-dependent transceptor. The standard definitions of transceptor are membrane proteins that possess both solute transport and receptor-like signalling activities or transporter-substrate complexes that transduce signals to the inside of a cell. None of these definitions apply to HAK5. What the data shows is that HAK5 transport activity is sensitive to substrate concentrations. This could be achieved either by allosteric regulation, as suggested for the bacterial homolog KimA, or feedback inhibition by K and activation by protons. As authors reference, this behaviour is reminiscent of K-dependent inactivation of Shaker K channels, which are not regarded as transceptors. To support the notion that HAK5 is a transceptor authors should link its activity to downstream signalling events beyond the fact that the cytoplasmic K concentration itself will trigger physiological and/or molecular responses.

2. Inactivation of HAK5 correlated positively with the K and H concentration in the bathing solution. Suppression of HAK5 inactivation in oocytes acidified with acetate (Fig 3D) led authors to conclude that neither the cytosolic pH nor the proton motive force drove HAK5 inactivation. However, cancelling the proton-motive force should reduce the rate of K/H symport and thus it is unclear whether HAK5 senses the external K concentration or the intracellular concentration of transported K or H to inactivate.

3. Mutation of residue Y450 in the so-called lower K binding site counteracted HAK5 inactivation, but it also reduced the transport rate by 90% making it unclear again what HAK5 senses for inactivation.

4. There seems to be some inconsistency of transport rates shown in Fig 1. The box plot in panel 1C shows an average current of 550 nA for sample HAK5+CIPK23/CBL1 with no outliers. However, the single current traces of HAK5+CIPK23/CBL1 shown in panel 1B depict a peak current of >900 nA and the current intensity in panel D was only 100 nA in 2 mM K, far from the 550 nA average in panel C. These values would be outliers in panel 1C.

5. Figures 3D and 5A,C show extremely fast HAK5 inactivation. However, this is not observed in Fig 1D. Why is that?

6. Could the rate of K-dependent inactivation of mutant Y450A be quantified, similar to Fig 1E, for comparison to the WT?

7. Please revise the correspondence between Fig 1B,C and sentence in lines 94-95. The plot shows HAK5 transporting in the presence of 2 mM K when co-expressed with CBL1/CIPK23.

8. I am surprised that pHluorin was fused to the N-terminus of HAK5 and not to the C-terminus, which would have been the first choice. Why this arrangement? Could authors show that the chimeric protein labelled the oocyte membrane as they did with the HAK5 dead mutants?

9. The definition of steady-state currents (I_{ss} , lines 195-196) should be done earlier when describing Fig 1E (lines 99-103).

10. X-axis is reversed in Fig 4 (plots B,C vs E)

11. Line 208, call to Fig 3C should be 3D?

Point-by-point reply to reviewers' comments.

Reviewer #1 (Remarks to the Author):

In this very interesting manuscript, the authors show that HAK5 from the plant *A. thaliana* is a hyperpolarization activated K⁺/H⁺ symporter, coupling the inward movement of potassium against its electrochemical gradient to the inwardly directed proton motive force (PMF). A detailed characterization of the transport mechanism that includes electrophysiology, fluorescence measurements, mutagenesis, as well as kinetic and structural modeling, reveals that the transporter is able to bind K⁺ with high affinity but also responds to high K⁺ concentrations by inactivating. While dependent on K⁺ concentration, the inactivation process appears to be independent of cytoplasmic pH or PMF. Ammonium is found to be transported by AtHAK5, but not as efficiently as K⁺. Nevertheless, at high concentrations, ammonium can outcompete K⁺ for transport. Furthermore, based on homology modeling with KimA from *B. subtilis*, tyrosine 450 is found to be critical to the high affinity binding of K⁺ to AtHAK5. Overall, the manuscript is clearly written, and the conclusions are well supported by the data.

We thank the reviewer for this positive feedback.

One point that needs to be addressed concerns the classification of AtHAK5 as a transceptor. In the manuscript, the word transceptor appears to be used as a synonym for high affinity transporter. While transceptors do have a high affinity site for the ion or molecule they transport, they also directly or indirectly affect downstream effectors through signaling cascades like receptors. In the literature cited by the authors for the definition of transceptor (refs. 36-37), intracellular signaling is mediated by phosphorylation in Ho et al. Cell 2009, or by second messengers, Ca²⁺ or H⁺, in Dreyer et al. iScience 2022. In their description of AtHAK5, the authors show that PMF is a driver for the movement of potassium against its electrochemical gradient. However, it is unclear whether protons serve the additional role of second messengers for AtHAK5. Are there pH-dependent cellular events that can be linked to the activity of AtHAK5 under low K⁺ conditions?

We have understood the referee's point, which is also shared by the other reviewers. We used the term "transceptor" in analogy to what was proposed for NRT1.1 (Ho et al. 2009; Ho and Tsay 2010; Gojon et al. 2011). The experimental data presented in this work indicate that AtHAK5 is not just a high-affinity transporter but has a mechanism that allows it to measure the external potassium concentration, essentially functioning as a true sensor of the external potassium concentration.

In the simplest form, a primary sensor reports on the level of a specific compound and couples the signaling function via a feedback loop to transport functionality. Feedback could be in the form of a conformational change within the sensor, which almost instantaneously modulates the uptake of that nutrient (Podar and Maathuis 2022).

According to this perspective, the K⁺-dependent activation and inactivation of AtHAK5 might reflect a nutrient sensor with a feed-back loop embedded in the structure. When sensing a drop in external K⁺ into the low μMolar range, AtHAK5 switches from its inactive state into its active state to maintain K⁺ uptake under low-K⁺ conditions. Under raising concentrations of external K⁺, HAK5 inhibits further transport activity to prevent excessive membrane depolarization and cytosolic acidification under conditions that allow K⁺ uptake via the hyperpolarization-activated shaker channel AKT1. Otherwise, the HAK5 mediated depolarization of the plasma membrane would counteract AKT1 activity until HAK5 is eventually inactivated by a protein phosphatase or degraded. This is reminiscent of the transceptor BOR1 (Yoshinari et al. 2021) that promotes own ubiquitination and degradation according to local B concentrations. Thus, B-sensing by the transceptor BOR1 is coupled with a feedback loop resulting in a reduction in B-transport.

Transceptors are membrane proteins that carry out both transport and signaling functions (Podar and Maathuis 2022) as it was shown for the human lysosomal amino acid transporter SLC38A9 (Scalise et al. 2019) or the amino acid transporter Gap1 from yeast (Kriel et al. 2011; Diallinas 2017). Downstream signaling is often linked to transport-dependent conformational changes, e.g. structural transition and protomer coupling of NRT1.1 are promoted by nitrate binding and were found essential for NRT1.1 function as sensor (Rashid et al. 2020; Rashid et al. 2018)

However, the referee is right that the link to intracellular signaling is scant. We show that the symport of K^+ and H^+ results in an acidification of the cytosol (Fig 3A). This increase in cytosolic protons could act as pH signal that is initiating downstream events. One possibility is that the pH signal directly activates the pH sensor SLAH3 (Lehmann et al. 2021). The activation of this anion channel would lead to the depolarization of the plasma membrane, thus translating the pH signal into an electrical signal, e.g. for long distance signaling.

Just recently, Dreyer et al (Dreyer et al. 2022) could demonstrate via a computational cell biology approach that the " K^+ -homeostat (the sum of all K^+ channels and transporters), in combination with the proton pump, could serve as a sensor for changes in $[K^+]_{apo}$. The external signal of a relative change in $[K^+]_{apo}$ was converted into a change in internal pH. Because regulation of cytosolic pH is coupled to cellular metabolism, the change in the extracellular nutrient concentration sensed by the K^+ -homeostat might thus directly influence cellular processes", thus reflecting transceptor-like characteristics. As AtHAK5 is the main player under K^+ starvation mediating K^+/H^+ symport and transport-mediated cytosolic acidification (Fig. 3A) AtHAK5 could be the key player in such a transceptor network.

However, we agree that future studies are needed to in detail investigate such putative downstream signaling relations and proof the transceptor function of AtHAK5, e.g. analyzing the transcriptome of *hak5* loss-of-function mutant plants could determine to what extend the transcriptional response to K-starvation is missing in the mutant, similar to what has been determined for the nitrate transporter and sensor NRT1.1/NPF6.3 (Ho et al. 2009; Maghiaoui et al. 2020).

We realize that there is a lack of terminology in the field to indicate this type of function and we agree that the link between K^+ sensing, regulation of transport activity and putative downstream signaling is still scant. Therefore, we tone down our statements and propose the term "transensor," which is a fusion of the words "transporter" and "sensor". Furthermore, we extended the discussion about the definition of nutrient transceptors and the classification of AtHAK5.

The authors should explain the K^+/H^+ stoichiometry they assumed in the kinetic modeling of transport. Also, it is not clear why they initially have a 6-state model in Fig. S2. Why is the value for the $k_{2,1}$ not provided in supplemental table 1?

The six-state model with a K^+/H^+ stoichiometry of 1:1 is the mathematical model commonly proposed to describe the secondary active transport of potassium coupled to protons in algae and plants, as demonstrated, for example, by the work of Blatt et al., 1987 ((Blatt et al. 1987), see also (Rodríguez-Navarro 2000) and references therein). Six states are required: the state without any bound ions in which the transporter exposes empty binding sites for protons and potassium towards the extracellular side (C1), the C2 state with a proton bound on the extracellular side, the C3 state with proton and potassium bound on the extracellular side, and the C4 state reached by a conformational change of the transporter. In C4, the proton and potassium face the cytosolic side. When potassium is released into the cytosol in C4, the state with only a proton bound on the cytosolic side (C5) is obtained. Potassium release into the cytosol leads to C6, the empty transporter with binding sites for protons and potassium facing the cytosolic side. A further conformational change leads from C6 to C1. We have added some of these details in "Methods details" in the "Kinetic model for HAK5-mediated peak currents" section.

To reduce the number of free parameters, since we are interested in the effects of extracellular potassium and protons, we group states C4, C5, and C6 into a single state (C4), resulting in a four-state model. The parameter k_{21} is not a free parameter but is obtained from equation (3) as a consequence of reversibility principle. This detail, along with the values of K_{12} in the WT and mutant transporter, has been added to the legend of Table 1.

Why was a Boltzmann function used to fit the data in Fig 3B, instead of a Hill equation? The methods section states that the Hill equation was used to calculate the pH-dependent half maximal activity of AtHAK5. If that is the case, the value of the Hill coefficient derived from the fit should be provided as well.

The referee is right, the information in the text is wrong. As stated in the figure legend and methods section, the pH-dependent half maximal activity was fitted with a Hill equation. We changed the information in the revised manuscript. The coefficients are now provided in the figures or in the respective figure legend: Hill coefficient HAK5 WT (H^+) = 1.67 ± 0.3 ($200 \mu M K^+$) and 2.0 ± 0.3 ($20 \mu M K^+$); Y450A = 2.17 ± 0.5

The Michaelis-Menten equation used to calculate I_{peak} in the methods section seems to be missing a current term. It could only be correct the way it is written now if I_{peak} were somehow normalized and rendered dimensionless.

The peak currents are effectively normalized to the value of the evoked current at $-120 mV$, thus they are dimensionless. For the sake of clarity, we have added the term I_{max_norm} to the equation provided in the text. The value of I_{max_norm} obtained from the fitting procedure is practically equal to 1 (as we had assumed in the equation of the previous version).

We have also corrected the value of K_m ; by mistake, it had remained a value obtained from preliminary data.

Minor points

On page 3, akt1-1 is not introduced.

We now introduce akt1-1 earlier.

It is recommended to maintain consistency about axis scales. In Fig.2B, the scale of the x-axis is not logarithmic whereas it is logarithmic in equivalent graphs in all other figures.

The reviewer is right, we changed the scale to logarithmic in all figures.

When the authors write “opus operandi”, they probably mean “modus operandi.”

We changed it to “modus operandi”

The word “blotted” is used in multiple places in the figure legends. It should be replaced with “plotted.”

Thank you, we replaced it by “plotted”

Reviewer #2 (Remarks to the Author):

In their study, Maierhofer et al. present the biophysical properties of AtHAK5, a member of the KUP/HAK/KT family of potassium transporters, by means of heterologous expression in *Xenopus*

oocytes and two-electrode voltage clamp. They found the hallmark properties of a high-affinity K⁺/H⁺ symporter. An interesting feature is the presence of an inactivation process at elevated K⁺ concentrations, at which apparently a K⁺-selective ion channel takes over in vivo to accomplish root K⁺ uptake. The molecular mechanisms of symport and inactivation were investigated using kinetic modeling, structural modeling and site-directed mutagenesis.

This is a solid biophysical study going deep into the molecular mechanisms of AtHAK5 function. The manuscript is for the most part clearly written and presented. Some points however require to be clarified or corrected.

We thank the reviewer for this positive feedback.

1. Much weight is laid on the definition of AtHAK5 as a K⁺-sensing transceptor. Is this justified? It is clear that every transport protein in some way “senses” its substrate. Here, high external K⁺ concentrations trigger an intrinsic inactivation process in the same transport protein, which may make sense from a physiological and energetical point of view. However, there is no indication that this information on the K⁺ status is transmitted to the cell and influences secondary processes.

We have understood the referee's point, which is also shared by the other reviewers. We used the term “transceptor” in analogy to what was proposed for NRT1.1 (Ho et al. 2009; Ho and Tsay 2010; Gojon et al. 2011). The experimental data presented in this work indicate that AtHAK5 is not just a high-affinity transporter but has a mechanism that allows it to measure the external potassium concentration, essentially functioning as a true sensor of the external potassium concentration.

In the simplest form, a primary sensor reports on the level of a specific compound and couples the signaling function via a feedback loop to transport functionality. Feedback could be in the form of a conformational change within the sensor, which almost instantaneously modulates the uptake of that nutrient (Podar and Maathuis 2022).

According to this perspective, the K⁺-dependent activation and inactivation of AtHAK5 might reflect a nutrient sensor with a feed-back loop embedded in the structure. When sensing a drop in external K⁺ into the low μMolar range, AtHAK5 switches from its inactive state into its active state to maintain K⁺ uptake under low-K⁺ conditions. Under raising concentrations of external K⁺, HAK5 inhibits further transport activity to prevent excessive membrane depolarization and cytosolic acidification under conditions that allow K⁺ uptake via the hyperpolarization-activated shaker channel AKT1. Otherwise, the HAK5 mediated depolarization of the plasma membrane would counteract AKT1 activity until HAK5 is eventually inactivated by a protein phosphatase or degraded. This is reminiscent of the transceptor BOR1 (Yoshinari et al. 2021) that promotes own ubiquitination and degradation according to local B concentrations. Thus, B-sensing by the transceptor BOR1 is coupled with a feedback loop resulting in a reduction in B-transport.

Transceptors are membrane proteins that carry out both transport and signaling functions (Podar and Maathuis 2022) as it was shown for the human lysosomal amino acid transporter SLC38A9 (Scalise et al. 2019) or the amino acid transporter Gap1 from yeast (Kriel et al. 2011; Diallinas 2017). Downstream signaling is often linked to transport-dependent conformational changes, e.g. structural transition and protomer coupling of NRT1.1 are promoted by nitrate binding and were found essential for NRT1.1 function as sensor (Rashid et al. 2020; Rashid et al. 2018)

However, the referee is right that the link to intracellular signaling is scant. We show that the symport of K⁺ and H⁺ results in an acidification of the cytosol (Fig 3A). This increase in cytosolic protons could act as pH signal that is initiating downstream events. One possibility is that the pH signal directly activates the pH sensor SLAH3 (Lehmann et al. 2021). The activation of this anion channel would lead to the depolarization of the plasma membrane, thus translating the pH signal into an electrical signal, e.g. for long distance signaling.

Just recently, Dreyer et al (Dreyer et al. 2022) could demonstrate via a computational cell biology approach that the “K⁺-homeostat (the sum of all K⁺ channels and transporters), in

combination with the proton pump, could serve as a sensor for changes in $[K^+]_{apo}$. The external signal of a relative change in $[K^+]_{apo}$ was converted into a change in internal pH. Because regulation of cytosolic pH is coupled to cellular metabolism, the change in the extracellular nutrient concentration sensed by the K^+ -homeostat might thus directly influence cellular processes", thus reflecting transceptor-like characteristics. As AtHAK5 is the main player under K^+ starvation mediating K^+/H^+ symport and transport-mediated cytosolic acidification (Fig. 3A) AtHAK5 could be the key player in such a transceptor network.

However, we agree that future studies are needed to in detail investigate such putative downstream signaling relations and proof the transceptor function of AtHAK5, e.g. analyzing the transcriptome of *hak5* loss-of-function mutant plants could determine to what extend the transcriptional response to K-starvation is missing in the mutant, similar to what has been determined for the nitrate transporter and sensor NRT1.1/NPF6.3 (Ho et al. 2009; Maghiaoui et al. 2020).

We realize that there is a lack of terminology in the field to indicate this type of function and we agree that the link between K^+ sensing, regulation of transport activity and putative downstream signaling is still scant. Therefore, we tone down our statements and propose the term "transensor," which is a fusion of the words "transporter" and "sensor". Furthermore, we extended the discussion about the definition of nutrient transceptors and the classification of AtHAK5.

2. The Y450A mutant showed an interrelation between K^+ and H^+ binding, also supported by the kinetic model. This is an interesting finding, but was not explained further. How could this work mechanistically?

The apparent affinity of the transporter to protons is described by the model through equation 17 and is a complex function of all the free parameters of the model. In particular, the constant K_{023} , crucial in modulating the apparent affinity of the transporter to potassium, as shown by equation 16, appears in equation 17 both in the numerator and denominator. Consequently, it does not seem possible to provide a mechanistic interpretation based on the kinetic model. However, the prediction of the correct interrelationship, at least qualitatively, between K^+ and H^+ binding in the wild-type transporter and the mutant is a demonstration of the validity of the model, as rightly pointed out by the referee. Also, upon the basis of the theoretical structure model of HAK5 and the functional data derived from the single amino acid mutations, a mechanistic interpretation of a potential interrelationship between the potassium and proton affinity/binding is very difficult. However, if we speculate on the effect of mutating tyrosine 450 to alanine, we can assume that the accessibility for water is increased when the bulky side chain of Y450 is replaced with the much smaller alanine. Hence, dehydration of potassium which is likely a prerequisite for K^+ transport is possibly impaired resulting in a lower affinity for potassium. At the same time, the increased access of water to cation binding site might facilitate the proton transport into the pore region of the ion channel. Thus, increased water access in the variant Y450A might affect K^+ and H^+ binding in a reciprocal manner.

3. Does the Y450A exchange (Figure 6) equally affect the other cation substrates (Figures 2, S1) in terms of affinity and inactivation?

We thank the referee for this question and performed the requested experiments. The results are provided in the new Supp Fig S6 of the revised manuscript.

Thereby, the selectivity of Y450A was found slightly different compared to the WT (new Suppl Fig S6A). Whereas transport of K^+ and Rb^+ was comparable alike in the WT (Fig 2C, Supp Fig S2A), Cs^+ and NH_4^+ elicited currents were reduced compared to the WT (new Suppl Fig S6A). However, in Y450A the

affinity for all tested cations is strongly shifted to higher concentrations, thus into the millimolar range (new Supp Fig S6B, C and D).

Similar to the WT, inactivation in the mutant Y450A occurs in external cation concentrations higher than K_m . Thus, inactivation of Y450A in K^+ and Rb^+ starts in 10 mM (K^+) or 20 mM (Rb^+), respectively. However, no inactivation was observed in the presence of NH_4^+ or Cs^+ in the tested concentration range.

Supplemental Figure S6: Related to Fig.6, Cation dependency of AtHAK5 mutant Y450A.

A) Cation-induced currents response at -120 mV of oocytes co-expressing AtHAK5 mutant Y450A and CIPK23/CBL1 in the presence of either 100 mM Li^+ , Na^+ , K^+ , Rb^+ , Cs^+ or NH_4^+ . Left panel: Representative current trace is shown. Right panel: Box plot of cation-induced currents at -120 mV ($n = 5$ experiments). **B)**

Left panel: Representative current response of AtHAK5 Y450A/CIPK23/CBL1 co-expressing oocytes upon application of different Rb⁺ concentrations. Right panel: Normalized whole-oocyte Rb⁺-induced peak currents (ΔI_{peak}) at -120 mV (pH4.5) plotted against the applied Rb⁺-concentration. K_m (Rb⁺) was calculated by fitting ΔI_{peak} with a Michaelis-Menten equation. (n = 5 experiments, mean \pm SD). **C)** Left panel: Representative current response of AtHAK5 Y450A/CIPK23/CBL1 co-expressing oocytes upon application of different NH₄⁺ concentrations. Right panel: Normalized whole-oocyte NH₄⁺-induced peak currents (ΔI_{peak}) at -120 mV (pH4.5) plotted against the applied NH₄⁺-concentration. K_m (NH₄⁺) was calculated by fitting ΔI_{peak} with a Michaelis-Menten equation. (n = 5 experiments, mean \pm SD). **D)** Left panel: Representative current response of AtHAK5 Y450A/CIPK23/CBL1 co-expressing oocytes upon application of different Cs⁺ concentrations. Right panel: Normalized whole-oocyte Cs⁺-induced peak currents (ΔI_{peak}) at -120 mV (pH4.5) plotted against the applied Cs⁺-concentration. K_m (Cs⁺) was calculated by fitting ΔI_{peak} with a Michaelis-Menten equation. (n = 5 experiments, mean \pm SD).

4. The pH dependence experiments (Figure 6D) was performed at 200 micromolar K⁺ for the WT (corresponding to about 10 times the K_m for K⁺) and 10 mM for the Y450A mutant (corresponding to the K_m for K⁺). Can these different conditions be compared?

We thank the reviewer for this comment. When measuring the pH dependency in 20 μM K⁺, K_m (H⁺) was similar compared to 200 μM K⁺. Thus, K_m (H⁺) seems to be K⁺-independent and the K⁺ concentration has no influence on our analysis shown in Fig 6D. This AtHAK5 characteristic is now described in the revised manuscript and shown in new Supp Fig S3B and C.

Supplemental Figure S3: Related to Fig.3, pH-dependency of AtHAK5.

B) Representative current response of AtHAK5/CIPK23/CBL1 co-expressing oocytes upon application of 20 μM K⁺ at different pH (as indicated in the figure). **C)** Whole-oocyte K⁺-induced peak currents (ΔI_{peak}) at -120 mV plotted against the applied H⁺-concentration. K_m (H⁺) was calculated by fitting ΔI_{peak} with a Hill equation. (n = 4 experiments, mean \pm SD).

5. Lines 94/95: It is stated that in oocytes co-expressing AtHAK5 and CBL1 “no transport activity” could be recorded at 2 mM K⁺. But see Figure 1B! Please clarify!

The referee is right, our statement is misleading. After switching from 0 potassium to 2 mM K⁺ a strong transient current response can be observed with fast deactivation kinetics. However, after prolonged exposure to high K⁺, no transport activity can be recorded.

We clarified our statement by including an additional sentence describing the data shown in Fig 1B and C.

“Like AKT1, AtHAK5 transport activity strongly depends on its co-expression with CBL1 and CIPK23, since inward K⁺ currents increased tenfold in the presence of the Ca²⁺-sensor kinase pair compared to the current amplitudes observed for AtHAK5 expressed alone (Fig. 1B and C, c.f. (Scherzer et al. 2015))”.

6. Order of panels in Figure 2: the authors may consider to change panel C to A, in order to comply with the order in the text.

The referee is right, we changed the order of the panels.

7. It is stated that both Cs⁺ currents (line 108) and NH₄⁺ currents (line 115) were “about 40% smaller” than K⁺ currents. In Figure 2C, Cs⁺ currents and NH₄⁺ currents are not equal though. Please correct!

The reviewer is right. We now provide the exact relations in the revised manuscript with $51.1 \pm 4.4\%$ for Cs⁺ and $37.6 \pm 9.6\%$ for NH₄⁺.

8. The part on the pH dependence (lines 126-150) should be written more clearly. Convincing evidence for a K⁺/H⁺ symport mechanism comes from the concomitant cytosolic acidification during inward K⁺ transport (Figure 3C). Measurements at a fixed external (Figure 3A/B) or internal pH (Figure 3D) may simply indicate pH-dependent modulation.

The referee is right. We rearranged the figure and changed Fig3C to Fig3A to highlight the transport dependent acidification of the cytosol as proof for K⁺/H⁺ symport.

9. At several positions within the manuscript, please change “blotted against..” into “plotted against..”

Thank you, we replaced it by “plotted”

Reviewer #3 (Remarks to the Author):

This work shows quite elegantly that the HAK5 transport mechanism is K/H symport. HAK5 activity was maximal at μM K concentrations and decreased in the mM range in agreement with the external K concentration at which this transporter operates in arabidopsis roots. HAK5 was activated by membrane hyperpolarisation and by external acidic pH by a mechanism separate from the mere electrophoretic pull of protons. Authors document that ammonium is a physiologically relevant substrate of HAK5. Last, HAK5 transport inactivated in a K- and time-dependent manner, and residue Y450 involved in K binding was critical for this feature. Authors propose that HAK5 is a transceptor that operates as a low-K sensor.

We thank the reviewer for this positive feedback.

Comments ordered in decreasing order of importance:

1. Authors conclude that HAK5 is a K-dependent transceptor. The standard definitions of transceptor

are membrane proteins that possess both solute transport and receptor-like signalling activities or transporter-substrate complexes that transduce signals to the inside of a cell. None of these definitions apply to HAK5. What the data shows is that HAK5 transport activity is sensitive to substrate concentrations. This could be achieved either by allosteric regulation, as suggested for the bacterial homolog KimA, or feedback inhibition by K and activation by protons. As authors reference, this behaviour is reminiscent of K-dependent inactivation of Shaker K channels, which are not regarded as transceptors. To support the notion that HAK5 is a transceptor authors should link its activity to downstream signalling events beyond the fact that the cytoplasmic K concentration itself will trigger physiological and/or molecular responses.

We have understood the referee's point, which is also shared by the other reviewers. We used the term "transceptor" in analogy to what was proposed for NRT1.1 (Ho et al. 2009; Ho and Tsay 2010; Gojon et al. 2011). The experimental data presented in this work indicate that AtHAK5 is not just a high-affinity transporter but has a mechanism that allows it to measure the external potassium concentration, essentially functioning as a true sensor of the external potassium concentration.

As correctly mentioned by the referee, Shaker channels are not defined as transceptors so far. However, the K⁺ dependent inactivation of the animal Shaker channel KCNQ1 by high external K⁺ can be directly linked to downstream signaling (Abrahamyan et al. 2023). The authors found that: "K_o⁺ sensitivity of KCNQ1/KCNE1 under these conditions will be a powerful mechanism for regulation of K⁺ flow toward endolymph serving as an important feedback regulatory tool for maintenance of the high endolymph [K⁺]." "Since K⁺ diffusion through the apical membrane of MCs (marginal cells) contributes to the generation of the endocochlear potential, the K_o⁺-sensitivity of KCNQ1 is essential for hearing and balance." This characteristic would specify KCNQ1 as transceptor linking the sensing of external K⁺ and K⁺-dependent inactivation with a physiological response.

In the simplest form, a primary sensor reports on the level of a specific compound and couples the signaling function via a feedback loop to transport functionality. Feedback could be in the form of a conformational change within the sensor, which almost instantaneously modulates the uptake of that nutrient (Podar and Maathuis 2022).

According to this perspective, the K⁺-dependent activation and inactivation of AtHAK5 might reflect a nutrient sensor with a feed-back loop embedded in the structure. When sensing a drop in external K⁺ into the low μMolar range, AtHAK5 switches from its inactive state into its active state to maintain K⁺ uptake under low-K⁺ conditions. Under raising concentrations of external K⁺, HAK5 inhibits further transport activity to prevent excessive membrane depolarization and cytosolic acidification under conditions that allow K⁺ uptake via the hyperpolarization-activated shaker channel AKT1. Otherwise, the HAK5 mediated depolarization of the plasma membrane would counteract AKT1 activity until HAK5 is eventually inactivated by a protein phosphatase or degraded. This is reminiscent of the transceptor BOR1 (Yoshinari et al. 2021) that promotes own ubiquitination and degradation according to local B concentrations. Thus, B-sensing by the transceptor BOR1 is coupled with a feedback loop resulting in a reduction in B-transport.

Transceptors are membrane proteins that carry out both transport and signaling functions (Podar and Maathuis 2022) as it was shown for the human lysosomal amino acid transporter SLC38A9 (Scalise et al. 2019) or the amino acid transporter Gap1 from yeast (Kriel et al. 2011; Diallinas 2017). Downstream signaling is often linked to transport-dependent conformational changes, e.g. structural transition and protomer coupling of NRT1.1 are promoted by nitrate binding and were found essential for NRT1.1 function as sensor (Rashid et al. 2020; Rashid et al. 2018)

However, the referee is right that the link to intracellular signaling is scant. We show that the symport of K⁺ and H⁺ results in an acidification of the cytosol (Fig 3A). This increase in cytosolic protons could act as pH signal that is initiating downstream events. One possibility is that the pH signal directly activates the pH sensor SLAH3 (Lehmann et al. 2021). The activation of this anion channel would lead to the depolarization of the plasma membrane, thus translating the pH signal into an electrical signal, e.g. for long distance signaling.

Just recently, Dreyer et al (Dreyer et al. 2022) could demonstrate via a computational cell biology approach that the “K⁺-homeostat (the sum of all K⁺ channels and transporters), in combination with the proton pump, could serve as a sensor for changes in [K⁺]_{apo}. The external signal of a relative change in [K⁺]_{apo} was converted into a change in internal pH. Because regulation of cytosolic pH is coupled to cellular metabolism, the change in the extracellular nutrient concentration sensed by the K⁺-homeostat might thus directly influence cellular processes”, thus reflecting transceptor-like characteristics. As AtHAK5 is the main player under K⁺ starvation mediating K⁺/H⁺ symport and transport-mediated cytosolic acidification (Fig. 3A) AtHAK5 could be the key player in such a transceptor network.

However, we agree that future studies are needed to in detail investigate such putative downstream signaling relations and proof the transceptor function of AtHAK5, e.g. analyzing the transcriptome of *hak5* loss-of-function mutant plants could determine to what extent the transcriptional response to K-starvation is missing in the mutant, similar to what has been determined for the nitrate transporter and sensor NRT1.1/NPF6.3 (Ho et al. 2009; Maghiaoui et al. 2020).

We realize that there is a lack of terminology in the field to indicate this type of function and we agree that the link between K⁺ sensing, regulation of transport activity and putative downstream signaling is still scant. Therefore, we tone down our statements and propose the term "transensor," which is a fusion of the words "transporter" and "sensor". Furthermore, we extended the discussion about the definition of nutrient transceptors and the classification of AtHAK5.

2. Inactivation of HAK5 correlated positively with the K and H concentration in the bathing solution. Suppression of HAK5 inactivation in oocytes acidified with acetate (Fig 3D) led authors to conclude that neither the cytosolic pH nor the proton motive force drove HAK5 inactivation. However, cancelling the proton-motive force should reduce the rate of K/H symport and thus it is unclear whether HAK5 senses the external K concentration or the intracellular concentration of transported K or H to inactivate.

The referee is right that HAK5 mediated K⁺/H⁺-symport depends on the respective ion gradients. However, regarding inactivation properties we found that when we strongly acidified the cytosol by acetate, we could still observe high K⁺-dependent inactivation of WT HAK5 (Fig 3D), showing that inactivation is independent of the cytosolic H⁺-concentration. In contrast, with lower H⁺ gradients across the membrane (higher external pH) inactivation even increased (Fig 5C and D). Furthermore, as shown in Fig 1D, prolonged incubation in high potassium (2 mM) keeps HAK5 in an inactive state as long as high potassium is present. To clarify this point we provide a new dataset showing that after fast activation and deactivation in response to 2 mM K⁺, AtHAK5 is locked in the inactive state even in the absence of HAK5 mediated K⁺/H⁺-symport and thus in the absence of decreased cytosolic pH. Only when external K⁺ is reduced to low levels HAK5 activates and maintains K⁺/H⁺-symport without inactivation although cytosolic pH decreases (Fig 3A, new Supp. Fig 1A and B). This strongly supports our notion that HAK5 senses the external potassium concentration rather than cytosolic K⁺ or H⁺.

Supplemental Figure S1: Related to Fig.1, Potassium dependent activation of AtHAK5.

A) Potassium-induced current response at -120 mV of oocytes co-expressing AtHAK5 and CIPK23/CBL1 in the presence of different K⁺ concentrations at pH 4.5. Representative current trace is shown. **B)** Whole-oocyte low K⁺-induced peak currents (ΔI_{peak}) at -120 mV and pH4.5 from oocytes expressing AtHAK5 with CIPK23/CBL1 plotted against the applied K⁺ concentration ($n \geq 4$ experiments, mean \pm SD).

3. Mutation of residue Y450 in the so-called lower K binding site counteracted HAK5 inactivation, but it also reduced the transport rate by 90% making it unclear again what HAK5 senses for inactivation.

This is true in 2 mM K⁺. However, the potassium sensitivity of Y450A is strongly shifted to higher potassium concentrations. In 100 mM potassium current amplitude of Y450A is similar to WT HAK5 at 2 mM potassium (Fig S3C). Similar to WT HAK5, the mutant Y450A inactivates at K⁺-concentrations higher than K_m (K⁺) (Fig S3D)

4. There seems to be some inconsistency of transport rates shown in Fig 1. The box plot in panel 1C shows an average current of 550 nA for sample HAK5+CIPK23/CBL1 with no outliers. However, the single current traces of HAK5+CIPK23/CBL1 shown in panel 1B depict a peak current of >900 nA and the current intensity in panel D was only 100 nA in 2 mM K, far from the 550 nA average in panel C. These values would be outliers in panel 1C.

Figure 1B shows the original current traces without baseline subtraction, whereas 1C shows the delta I peak currents representing the potassium induced increase in current amplitude. As baseline currents in this batch of oocytes were about -150 μM, currents in Fig 1C are smaller than currents shown in Fig 1B. Different batches of oocytes may show different expression levels and different maximal HAK5 mediated current amplitudes. Because Fig 1D shows results from a different oocyte batch as Fig 1B and C, current amplitudes are different.

5. Figures 3D and 5A,C show extremely fast HAK5 inactivation. However, this is not observed in Fig 1D. Why is that?

Fig 1D shows activation of HAK5 after inactivation. When starting in 2mM K^+ HAK5 is in its inactive state. However, after switching to low potassium (20 μ M) HAK5 activates as inactivation of HAK5 only occurs at K^+ concentrations higher than K_m (K^+) (Figure 1E). This fact is also analysed in more detail in new Suppl. Fig 1A and B. The lack of K^+ -dependent inactivation at low K^+ is also shown in Figure 5A and C.

6. Could the rate of K-dependent inactivation of mutant Y450A be quantified, similar to Fig 1E, for comparison to the WT?

We thank the reviewer for this suggestion. We now provide the respective dataset in new Suppl. Fig S5D, E and F.

Similar to the WT, the mutant Y450A inactivates in a K^+ dependent manner (new Suppl. Fig S5D and E). When plotting the resulting steady-state currents against the K^+ concentration the maximum peak current is at K_m [K^+] (new Suppl. Fig S5F), reminiscent of the WT.

Supplemental Figure S5: Related to Fig.6, Molecular nature of K^+ sensing.

D) Representative current trace of oocytes co-expressing AtHAK5 Y450A and CIPK23/CBL1 at -120 mV and pH4.5 when challenged with different K^+ concentrations (as indicated in the figure). **E)** Degree of inactivation (in %) derived from similar experiments as shown in D) were plotted against the applied K^+ concentration (n = 5 experiments, mean \pm SD). **F)** Normalized whole-oocyte K^+ -induced peak currents (ΔI_{peak}) or steady-state currents (ΔI_{SS}) at -120 mV at pH4.5 are plotted against the applied K^+ -concentration.

K_m (K^+) was calculated by fitting ΔI_{peak} with a Michaelis-Menten equation. The modified Michaelis-Menten function used to fit ΔI_{SS} is described in the methods section ($n = 5$ experiments \pm SD).

7. Please revise the correspondence between Fig 1B,C and sentence in lines 94-95. The plot shows HAK5 transporting in the presence of 2 mM K when co-expressed with CBL1/CIPK23.

The referee is right, our statement is misleading. After switching from 0 potassium to 2 mM K^+ a strong transient current response can be observed with fast deactivation kinetics. However, after prolonged exposure to high K^+ , no transport activity can be recorded.

We clarified our statement by including an additional sentence describing the data shown in Fig 1B and C.

“Like AKT1, AtHAK5 transport activity strongly depends on its co-expression with CBL1 and CIPK23, since inward K^+ currents increased tenfold in the presence of the Ca^{2+} -sensor kinase pair compared to the current amplitudes observed for AtHAK5 expressed alone (Fig. 1B and C, c.f. (Scherzer et al. 2015))”.

8. I am surprised that pHluorin was fused to the N-terminus of HAK5 and not to the C-terminus, which would have been the first choice. Why this arrangement? Could authors show that the chimeric protein labelled the oocyte membrane as they did with the HAK5 dead mutants?

Suppl. Fig S5B shows oocytes expressing HAK5 WT and the different mutants tagged with YFP. Also in this experiment, HAK5 was N-terminally fused to YFP. In our hands the C-terminal fusion to YFP showed lower expression in oocytes. Therefore, pHluorin was also fused to the N-terminus of HAK5. The HAK5 WT-like transport function of oocytes expressing the chimeric protein (Fig 3C) strongly supports the membrane-localization. A representative image showing the pHluorin-labelling of the oocyte is now shown in new Supplemental Figure 3A in the revised manuscript.

A

Supplemental Figure S3: Related to Fig.3, pH-dependency of AtHAK5.

A) Representative image of an oocyte expressing WT HAK5 fused with pHluorin at the N-terminus compared to a control oocyte. Images of one quarter of an oocyte from at least four independent experiments are shown.

9. The definition of steady-state currents (I_{ss} , lines 195-196) should be done earlier when describing Fig 1E (lines 99-103).

In the text, the steady-state currents are mentioned and described in lines 195-196 for the first time, therefore we included the definition at this point. In lines 99-103 we only focus on the K^+ dependency of the peak currents (I_{peak}). Therefore, we think that the definition of I_{ss} at this point could be confusing without describing the respective data. However, the definition of I_{ss} is included in the figure legend of Fig 1E, so we hope that this is sufficient at this point.

10. X-axis is reversed in Fig 4 (plots B,C vs E)

We reversed the X-axis in Fig 4E.

11. Line 208, call to Fig 3C should be 3D?

The reviewer is right, we corrected to 3D.

Abrahamyan A, Eldstrom J, Sahakyan H, Karagulyan N, Mkrtchyan L, Karapetyan T, Sargsyan E, Kneussel M, Nazaryan K, Schwarz JR, Fedida D, Vardanyan V (2023) Mechanism of external K^+ sensitivity of KCNQ1 channels. *J Gen Physiol* 155 (5):e202213205.

doi:10.1085/jgp.202213205

Blatt MR, Rodriguez-Navarro A, Slayman CL (1987) Potassium-proton symport in *Neurospora*: kinetic control by pH and membrane potential. *J Membr Biol* 98 (2):169-189. doi:10.1007/BF01872129

Diallinas G (2017) Transceptors as a functional link of transporters and receptors. *Microb Cell* 4 (3):69-73. doi:10.15698/mic2017.03.560

Dreyer I, Li K, Riedelsberger J, Hedrich R, Konrad KR, Michard E (2022) Transporter networks can serve plant cells as nutrient sensors and mimic transceptor-like behavior. *iScience* 25 (4):104078. doi:10.1016/j.isci.2022.104078

Gojon A, Krouk G, Perrine-Walker F, Laugier E (2011) Nitrate transceptor(s) in plants. *Journal of Experimental Botany* 62 (7):2299-2308. doi:10.1093/jxb/erq419

Ho CH, Lin SH, Hu HC, Tsay YF (2009) CHL1 functions as a nitrate sensor in plants. *Cell* 138 (6):1184-1194. doi:10.1016/j.cell.2009.07.004

Ho CH, Tsay YF (2010) Nitrate, ammonium, and potassium sensing and signaling. *Curr Opin Plant Biol* 13 (5):604-610. doi:10.1016/j.pbi.2010.08.005

Kriel J, Haesendonckx S, Rubio-Teixeira M, Van Zeebroeck G, Thevelein JM (2011) From transporter to transceptor: signaling from transporters provokes re-evaluation of complex trafficking and regulatory controls: endocytic internalization and intracellular trafficking of nutrient transceptors may, at least in part, be governed by their signaling function. *Bioessays* 33 (11):870-879. doi:10.1002/bies.201100100

Lehmann J, Jorgensen ME, Fratz S, Muller HM, Kusch J, Scherzer S, Navarro-Retamal C, Mayer D, Bohm J, Konrad KR, Terpitz U, Dreyer I, Mueller TD, Sauer M, Hedrich R,

- Geiger D, Maierhofer T (2021) Acidosis-induced activation of anion channel SLAH3 in the flooding-related stress response of Arabidopsis. *Curr Biol* 31 (16):3575-3585 e3579. doi:10.1016/j.cub.2021.06.018
- Maghiaoui A, Gojon A, Bach L (2020) NRT1.1-centered nitrate signaling in plants. *J Exp Bot* 71 (20):6226-6237. doi:10.1093/jxb/eraa361
- Podar D, Maathuis FJM (2022) Primary nutrient sensors in plants. *iScience* 25 (4):104029. doi:<https://doi.org/10.1016/j.isci.2022.104029>
- Rashid M, Bera S, Banerjee M, Medvinsky AB, Sun GQ, Li BL, Sljoka A, Chakraborty A (2020) Feedforward Control of Plant Nitrate Transporter NRT1.1 Biphasic Adaptive Activity. *Biophys J* 118 (4):898-908. doi:10.1016/j.bpj.2019.10.018
- Rashid M, Bera S, Medvinsky AB, Sun GQ, Li BL, Chakraborty A (2018) Adaptive Regulation of Nitrate Transporter NRT1.1 in Fluctuating Soil Nitrate Conditions. *iScience* 2:41-50. doi:10.1016/j.isci.2018.03.007
- Rodríguez-Navarro A (2000) Potassium transport in fungi and plants. *Biochimica et Biophysica Acta (BBA) - Reviews on Biomembranes* 1469 (1):1-30. doi:[https://doi.org/10.1016/S0304-4157\(99\)00013-1](https://doi.org/10.1016/S0304-4157(99)00013-1)
- Scalise M, Galluccio M, Pochini L, Cosco J, Trotta M, Rebsamen M, Superti-Furga G, Indiveri C (2019) Insights into the transport side of the human SLC38A9 transporter. *Biochimica et Biophysica Acta (BBA) - Biomembranes* 1861 (9):1558-1567. doi:<https://doi.org/10.1016/j.bbamem.2019.07.006>
- Scherzer S, Bohm J, Krol E, Shabala L, Kreuzer I, Larisch C, Bemm F, Al-Rasheid KA, Shabala S, Rennenberg H, Neher E, Hedrich R (2015) Calcium sensor kinase activates potassium uptake systems in gland cells of Venus flytraps. *Proc Natl Acad Sci U S A* 112 (23):7309-7314. doi:10.1073/pnas.1507810112
- Yoshinari A, Hosokawa T, Beier MP, Oshima K, Ogino Y, Hori C, Takasuka TE, Fukao Y, Fujiwara T, Takano J (2021) Transport-coupled ubiquitination of the borate transporter BOR1 for its boron-dependent degradation. *The Plant Cell* 33 (2):420-438. doi:10.1093/plcell/koaa020

REVIEWERS' COMMENTS

Reviewer #1 (Remarks to the Author):

My previous concerns have been addressed in the thoroughly revised manuscript. The authors have toned down the classification of AtHAK5 as a “transceptor”, opting for the more conservative definition of “transensor”. While future research will be required to identify potential effectors linked to AtHAK5 activity under low K⁺ conditions, the current conclusion about the ability of AtHAK5 to both sense and transport potassium is well supported by the data.

Reviewer #2

In the revised manuscript, the authors have satisfactorily responded to most of my previous concerns, including also new data responding to Points 3 and 4.

Nevertheless, the “transceptor” question remains. The authors now toned down their statement and instead proposed the term “transsensor”, stressing the fact that AtHAK5 is able to “sense” elevated K⁺ concentrations responding with its own inactivation. In my view, this is rather some kind of quibbling. The critical point here is that there is no strong indication for a link between the regulation of AtHAK5 transport activity and downstream K⁺ signaling. As also admitted by the authors themselves.

Furthermore, the authors extended the Discussion part on nutrient transceptors and the putative role of AtHAK5. The proposed mechanisms however remain speculative and vague and do not contain clues to the specificity of low K⁺ signaling. Also, the authors do not relate to previous work suggesting that low soil K⁺ conditions might be sensed intracellularly (Wang et al., Dev Cell 2021).

In summary, I feel this study does not have sufficient depth to sustain a significant leap forward in our understanding of low K⁺ sensing.

Point-by-point reply to reviewers' comments.

Reviewer #1 (Remarks to the Author):

My previous concerns have been addressed in the thoroughly revised manuscript. The authors have toned down the classification of AtHAK5 as a “tranceptor”, opting for the more conservative definition of “transensor”. While future research will be required to identify potential effectors linked to AtHAK5 activity under low K⁺ conditions, the current conclusion about the ability of AtHAK5 to both sense and transport potassium is well supported by the data.

Response: We thank the reviewer for this positive feedback.

Reviewer #2

In the revised manuscript, the authors have satisfactorily responded to most of my previous concerns, including also new data responding to Points 3 and 4.

Nevertheless, the “transceptor” question remains. The authors now toned down their statement and instead proposed the term “transsensor”, stressing the fact that AtHAK5 is able to “sense” elevated K⁺ concentrations responding with its own inactivation. In my view, this is rather some kind of quibbling. The critical point here is that there is no strong indication for a link between the regulation of AtHAK5 transport activity and downstream K⁺ signaling. As also admitted by the authors themselves.

Furthermore, the authors extended the Discussion part on nutrient transceptors and the putative role of AtHAK5. The proposed mechanisms however remain speculative and vague and do not contain clues to the specificity of low K⁺ signaling. Also, the authors do not relate to previous work suggesting that low soil K⁺ conditions might be sensed intracellularly (Wang et al., Dev Cell 2021).

In summary, I feel this study does not have sufficient depth to sustain a significant leap forward in our understanding of low K⁺ sensing.

Response: We thank the reviewer for this critical feedback. Our data clearly show that AtHAK5 can sense the external K⁺ concentration. Upon a drop in external K⁺ to low micromolar concentrations, AtHAK5 activates to maintain K⁺ uptake under these conditions. However, we agree with the reviewer that the link between K⁺ dependent HAK5 activity and downstream K⁺ signaling is not supported by our data. Therefore, we revised our manuscript to avoid stating that AtHAK5 is a K⁺ sensor or transsensor.

The study of Wang et al., 2021 is already mentioned in the introduction of our manuscript. The authors observed a rapid decline of cytosolic K⁺ initiating in less 60 s after onset of low K⁺ stress that was strictly confined to a niche of postmeristematic cells at the meristematic zone/ elongation zone junction. But how low soil K⁺ triggers this drop in K⁺ within specific root cells is not clear. HAK5 activates in response to low K⁺ within several seconds, hence one could speculate that AtHAK5 initiates this low K⁺ response in postmeristematic cells. Thus, both studies are not compulsory contradictory. Nevertheless, future studies must

show if AtHAK5 is able to function as sensor or transceptor for low soil K⁺. This perspective is still included in the discussion of the manuscript.